# Stepwise polarisation of developing bilayered epidermis is mediated by aPKC and E-cadherin in zebrafish

Prateek Arora, Shivali Dongre, Renuka Raman[†], Mahendra Sonawane*

Department of Biological Sciences, Tata Institute of Fundamental Research, Mumbai, India

**Abstract** The epidermis, a multilayered epithelium, surrounds and protects the vertebrate body. It develops from a bilayered epithelium formed of the outer periderm and underlying basal epidermis. How apicobasal polarity is established in the developing epidermis has remained poorly understood. We show that both the periderm and the basal epidermis exhibit polarised distribution of adherens junctions in zebrafish. aPKC, an apical polarity regulator, maintains the robustness of polarisation of E-cadherin- an adherens junction component- in the periderm. E-cadherin in one layer controls the localisation of E-cadherin in the second layer in a layer non-autonomous manner. Importantly, E-cadherin controls the localisation and levels of Lgl, a basolateral polarity regulator, in a layer autonomous as well non-autonomous manner. Since periderm formation from the enveloping layer precedes the formation of the basal epidermis, our analyses suggest that peridermal polarity, initiated by aPKC, is transduced in a stepwise manner by E-cadherin to the basal layer.

**\*For correspondence:**
mahendras@tifr.res.in

**Present address:** [†]Weill Cornell Medical College, New York, United States

**Competing interests:** The authors declare that no competing interests exist.

## Introduction

The metazoan body is covered by the epidermis, an epithelium having a major function in the maintenance of milieu interior. This epithelium is multi-layered or stratified in vertebrates. During embryogenesis, the mammalian epidermis begins as a single layer, becomes bilayered and eventually develops into a multi-layered and keratinised tissue (*Blanpain and Fuchs, 2009*; *Evans and Rutter, 1986*; *Koster and Roop, 2007*; *M'Boneko and Merker, 1988*; *Nakamura and Yasuda, 1979*; *Saathoff et al., 2004*; *Smart, 1970*). In zebrafish, the early bilayered epidermis is formed of an outer periderm, which differentiates from the enveloping layer (EVL), and inner basal epidermis that develops from the non-neuralised ectoderm (*Bakkers et al., 2002*; *Chang and Hwang, 2011*; *Kimmel et al., 1990*). This bilayered epithelium is present in the embryonic and larval stages but becomes multi-layered during metamorphosis (*Chang and Hwang, 2011*; *Le Guellec et al., 2004*; *Lee et al., 2014*).

Several lines of evidence indicate that the epidermis is polarised at the tissue level (*Brandner et al., 2010*; *Getsios et al., 2004*; *Muroyama and Lechler, 2012*; *Watt, 2001*). While such polarisation exists at multiple levels, it is more profound in terms of localisation of various junctional components. Tight junction components such as Cldn-1, ZO-1 and Occludin are localised differentially across layers in the mammalian epidermis (*Brandner, 2009*; *Brandner et al., 2002*; *Kirschner and Brandner, 2012*). Desmosomal cadherins, plakophillins, envoplakin and periplakin show differentiation dependent changes in expression levels across layers in the epidermis (*Getsios et al., 2004*). Besides, hemidesmosomes mediating the cell-matrix adhesions form only in the basal domain of basal epidermal cells (*Hieda et al., 1992*; *Owaribe et al., 1990*; *Sonawane et al., 2005*). However, the steps and mechanisms involved in polarisation of the different

developmental forms of the epidermis – monolayered, bilayered and multilayered - have remained elusive.

A simple epithelium has a well-defined apicobasal polarity; the side that faces the lumen or external environment is defined as the apical domain and the side that is in contact with the basement membrane and neighbouring cells is called the basolateral domain (*Rodriguez-Boulan and Macara, 2014*; *Tepass, 2012*). In vertebrates, the tight junctions demarcate the apical domain and thus separate the apical domain from the basolateral domain (*Rodriguez-Boulan and Macara, 2014*; *Tepass, 2012*). In addition, simple epithelia possess an apical zonula adherens, a tight belt of cadherin mediated adherens junctions. The formation and proper apical localisation of tight and adherens junctions marks an important step in polarisation of the epithelial sheet and is tightly controlled by the epithelial polarity pathways (*Laprise and Tepass, 2011*; *Rodriguez-Boulan and Macara, 2014*). At least four modules or pathways are important for establishment and maintenance of apicobasal polarity in the simple epithelia. Amongst these aPKC-Par3-Par6, Crb-Sdt-Patj control the apical polarity and Lgl-Dlg-Scrb, Yurt-Cora-Nrx IV-Na$^+$K$^+$ATPase regulate the basolateral polarity (*Bilder et al., 2000*; *Bilder and Perrimon, 2000*; *Kemphues et al., 1988*; *Laprise et al., 2009*; *Tepass et al., 1990*; *Wodarz et al., 1995*). The loss of *apkc* function results in the disruption of adherens junctions in *Drosophila* neuroectoderm (*Wodarz et al., 2000*), the intestine and retinae of zebrafish (*Horne-Badovinac et al., 2001*), and in the neuroepithelium of mice (*Imai et al., 2006*). The absence of *lgl* function results in mislocalisation of the adherens junctions and expansion of apical domain in multiple *Drosophila* epithelial tissues (*Bilder et al., 2000*; *Tanentzapf and Tepass, 2003*). Similar to adherens junctions, formation and localisation of tight junctions is also shown to be regulated by cell polarity regulators *apkc* and *lgl* (*Chalmers et al., 2005*; *Iden et al., 2012*; *Yamanaka et al., 2006*; *Yamanaka et al., 2003*).

In mouse epidermis, aPKC regulates the cell division plane and maintains the stem cells in a quiescent stage in the hair follicles (*Helfrich et al., 2007*; *Niessen et al., 2013*; *Osada et al., 2015*). The aPKC function is also required to build trans epithelial resistance in the basal keratinocytes cell culture (*Helfrich et al., 2007*). In the embryonic epidermis, *heart and soul (has)/apkc λ/ι* functions to control the elongation of microridges, actin based apical projections on the zebrafish peridermal cells (*Raman et al., 2016*). Amongst the basolateral components, *penner/lgl2* function is essential for the formation of hemidesmosomes in the basal epidermis and to promote the elongation of microridges in the periderm in zebrafish (*Raman et al., 2016*; *Sonawane et al., 2005*; *Sonawane et al., 2009*). Furthermore, *psoriasis/atp1b1a* - encoding a subunit of Na$^+$K$^+$ATPase – regulates localisation of Lgl as well as E-cadherin and functions as a cell non-autonomous tumour suppressor in hypotonic conditions (*Hatzold et al., 2016*). Despite these earlier attempts, the functions of aPKC and Lgl in regulation of epidermal polarity and adherens junction formation, specifically in the developing vertebrate epidermis, have remained poorly understood.

Adherens junctions contribute towards the polarisation of simple epithelia. The loss of *shotgun/de-cadherin* and *armadillo/β-catenin* function is known to perturb the polarity and architecture of epithelial tissues in *Drosophila* (*Müller and Wieschaus, 1996*; *Tanentzapf et al., 2000*; *Tepass et al., 1996*). In mouse, epidermis- specific E-cadherin knockout shows mislocalisation of aPKC and Cdc42 (*Tunggal et al., 2005*). Moreover, E-cadherin is also required for the formation of tight junctions and regulation of hemidesmosome formation in the epidermis of mouse and zebrafish, respectively (*Rübsam et al., 2017*; *Sonawane et al., 2009*; *Tunggal et al., 2005*). So far, it has remained unclear whether E-cadherin is localised in a polarised manner in the epidermal layers and whether it regulates cell polarity in the developing epidermis.

Here we show that in the zebrafish embryonic bilayered epidermis, the adherens junctions show a polarised distribution in each layer of the epidermis. Unexpectedly, we observe that in the periderm, E-cadherin localisation increases from the apical to the basal domain, while the basal epidermis exhibits a reverse trend. Thus, E-cadherin levels are highest at the interlayer junction. Cell polarity regulator aPKCλ/ι controls the levels of E-cadherin in epidermal cells. Importantly, aPKCλ/ι function is required to maintain robustness of the polarised distribution of E-cadherin in peridermal cells. We further demonstrate that E-cadherin in one layer influences E-cadherin and Lgl localisation in the juxtaposed layer. Thus, our data reveal that E-cadherin, by virtue of being a connector of the two layers, transduces polarity cues across the epithelium.

## Results

### Adherens junctions exhibit a polarised distribution within each layer of the bilayered epidermis with maximum levels at the inter-layer interface

It has been shown that the epidermis, a stratified epithelium, is polarised at the tissue level (*Brandner et al., 2010*; *Getsios et al., 2004*; *Muroyama and Lechler, 2012*; *Watt, 2001*). To address whether a simple bilayered epidermis in developing zebrafish embryos exhibits polarity with respect to adherens junctions, we performed immunostainings for E-cadherin (Cdh-1), a trans-membrane component of adherens junctions. To demarcate the apical domain, we used aPKC that marks the apical domain in the periderm and analysed the localisation of E-cadherin along the apicobasal axis. Analysis of successive confocal slices in z-stacks and immuno-histological sections revealed that the localisation of E-cadherin increases gradually along the apicobasal axis of the peridermal cells at 48 hours post-fertilisation (hpf; *Figure 1A1, B1*). To investigate this E-cadherin polarity better, we analysed embryos stained for transgenically driven Lyn-GFP, which marks the peridermal membrane, (*Haas and Gilmour, 2006*; *Sonal et al., 2014*) and E-cadherin by histology and those additionally stained for aPKC by confocal microscopy. These analyses further indicated that the E-cadherin localisation increased gradually along the apico-basal axis in the periderm at 48 hpf (*Figure 1 B2, B3, B4*, *Figure 1—figure supplement 1A*, *Figure 1—video 1*). We further sought to analyse the localisation of E-cadherin with respect to Lgl2. Immunolocalisation analyses by confocal microscopy as well as histology revealed that E-cadherin colocalised with Lgl, both in the periderm as well as in the basal epidermis (*Figure 1—figure supplement 1B,C,D*).

To confirm the polarised localisation, we estimated the membrane levels of E-cadherin in the epidermal cells by measuring the fluorescence intensities in successive confocal slices along the z-axis, at various developmental stages. We normalised the cell heights, fitted the data along the cell height using 'locfit' and interpolated the intensity values at comparative intermediate cell heights (see Materials and methods). The interpolated intensities at the chosen cell heights were then plotted as boxplots and compared across various developmental stages. This analysis revealed that the mean E-cadherin levels increase gradually from the apical to the basal domain, plateauing at the basal domain of the peridermal cells, at all stages analysed between 24 to 72hpf. Thus, E-cadherin is localised in a polarised manner along the apicobasal axis in peridermal cells (*Figure 1 A1, C1*; *Figure 1—figure supplement 2A*). In the basal epidermis, this distribution is reversed. E-cadherin levels were found to be higher at the apical domain and decreased gradually towards the basal domain (*Figure 1 A2, B2, B5, C2*; *Figure 1—figure supplement 2B*). During development, the mean E-cadherin intensity at each z-level increased in both the periderm and basal epidermis (*Figure 1 C1, C2*; *Figure 1—figure supplement 2*). A similar trend was observed for β-catenin, another component associated with the adherens junctions, in the periderm and basal epidermis (*Figure 1—figure supplement 3*). Thus, the localisation of adherens junction components seems to increase over time in the epidermis, especially from 48hpf to 72hpf. To verify the polarised localisation of E-cadherin further, we measured the levels of E-cadherin in live at 48hpf by expressing an mCherry tagged version of E-cadherin in the background of Lyn-GFP. Corroborating the findings from fixed samples, we observed a gradual increase in E-cadherin-mCherry levels across the apico-basal axis in the periderm, and a reverse trend in the basal epidermis (*Figure 1—figure supplement 4A,B*).

To conclude, E-cadherin and β-catenin localisation indicate that adherens junctions are present in a polarised manner in the zebrafish bilayered larval epidermis. The membrane levels of these two adherens junction components increase along the apicobasal axis in the periderm. This trend is reversed in the basal epidermis. Thus, the maximum levels of E-cadherin and β-catenin are present at the interface of both the layers. Besides, there is a significant increase in the membrane localisation of E-cadherin and β-catenin during development, especially between 48hpf to 72hpf, suggesting an increase in the formation of adherens junctions during epidermis development.

### aPKC promotes robustness of the polarisation of adherens junctions in the bilayered epidermis

In simple epithelia, polarity pathways Par3-Par6-aPKC and Dlg-Scrib-Lgl regulate the localisation of adherens junction components (*Bilder et al., 2000*; *Rodriguez-Boulan and Macara, 2014*;

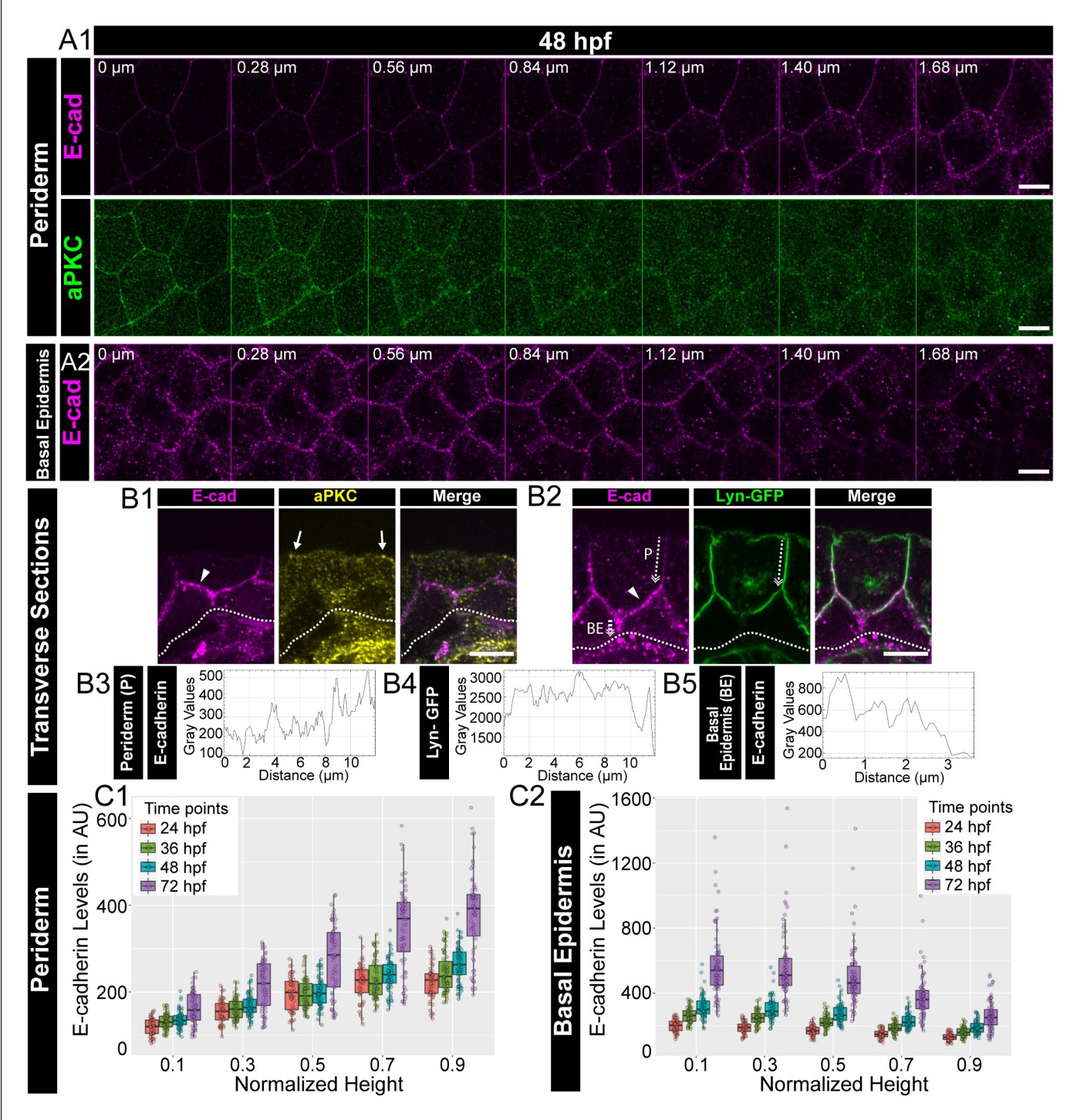

**Figure 1.** Polarised localisation of E-cadherin in zebrafish bilayered epidermis during early development. Confocal images of the periderm (**A1**) and basal epidermis (**A2**) showing localisation of E-cadherin (magenta) and aPKC (green) along various cell-heights at 48hpf in wild type embryos. Note 0 µm is the apical most section in both periderm as well as in the basal epidermis. Transverse sections of the epidermis covering the eye showing localisation of (**B1**) E-cadherin (magenta) and aPKC (yellow) and (**B2**) E-cadherin (magenta) and Lyn-GFP (green). Dotted arrows mark the cell boundary (**B2**) used for line intensity profiles for E-cadherin (**B3**) and Lyn-GFP (**B4**) in the Periderm (P) and for E-cadherin (**B5**) in Basal epidermis (BE). Quantification of E-cadherin levels along apicobasal axis at the normalised cell heights in the periderm (**C1**) and the basal epidermis (**C2**) at different developmental time points. Arrowheads in B1 and B2 point to maximum levels of E-cadherin at the interface of the two layers. Solid arrows in B1 point to aPKC localisation. Dotted lines mark the base of the epidermis. Scale bar is equivalent to 10 µm (**A1, A2, B1, B2**). AU = Arbitrary Units. Source file with fluorescence intensities for periderm and basal epidermis is available as *Figure 1—source data 1* and *2* respectively.

*Figure 1 continued on next page*

*Figure 1 continued*

The online version of this article includes the following video, source data, and figure supplement(s) for figure 1:

**Source data 1.** Fluorescence intensities of E-cadherin localisation in the periderm of wild type embryos at different deveoplmental time points.

**Source data 2.** Fluorescence intensities of E-cadherin localisation in the basal epidermis of wild type embryos at different deveoplmental time points.

**Figure supplement 1.** Localisation of E-cadherin with cell polarity markers in the zebrafish embryonic epidermis.

**Figure supplement 2.** E-cadherin localisation is polarised during early epidermis development.

**Figure supplement 3.** Localisation of β-catenin in the periderm and basal epidermis indicates the presence of adherens junctions during the epidermis development.

**Figure supplement 4.** Live imaging analysis confirms that E-cadherin localises in a polarised manner in the early developing epidermis.

**Figure supplement 4—source data 1.** Fluorescence intensities of E-cadherin-mCherry in the periderm of live wild type embryos in the background of Lyn-GFP at 48hpf.

**Figure supplement 4—source data 2.** Fluorescence intensities of E-cadherin-mCherry in the basal epidermis of live wild type embryos in the background of Lyn-GFP at 48hpf.

**Figure 1—video 1.** Localisation of E-cadherin and aPKC in the epidermal bilayer.

https://elifesciences.org/articles/49064#fig1video1

*Wodarz et al., 1995*; *Wodarz et al., 2000*). We asked whether aPKC and Lgl - components of the apical and basolateral polarity pathways respectively, regulate the polarised distribution of adherens junctions in the bilayered epidermis. While aPKC localises to the apical domain of the periderm, Lgl is localised to the basolateral domain of the peridermal cells and to the apicolateral domain of the basal epidermal cells (*Figure 1 A1, B1*, *Figure 1—figure supplement 1*, *Figure 2—figure supplement 1*, *Raman et al. (2016)*; *Sonawane et al. (2009)*). To check the importance of these two components in adherens junction formation, we analysed *heart and soul* (*has/apkc*) and *penner* (*pen/lgl2*) mutants (*Horne-Badovinac et al., 2001*; *Sonawane et al., 2005*). We observed a decrease in E-cadherin levels in the periderm as well as in the basal epidermis of *has/apkc* mutant embryos at 48hpf (*Figure 2 A1, B1*). Co-staining with Lyn-GFP marker and sectioning confirmed the reduction in E-cadherin levels (*Figure 2—figure supplement 2A,B*). However, this phenotype varied - from strong to mild to nil - in mutant embryos obtained from different pairs.

On comparing the cell morphology between *has/apkc* mutants and wild-type siblings, we found that mutants had significantly higher number of taller peridermal cells as compared to the siblings (*Figure 2 A2*, Mann Whitney U test, p<0.05). The apical perimeter of the mutant peridermal cells was also smaller than that of siblings (*Figure 2 A3*, Mann Whitney U test, p<0.05). We probed if the abnormal cellular morphology was restricted only to the taller cells. Taking the distribution of the sibling cells as a reference, the cells were classified as tall, medium or short (see Materials and methods for details; *Figure 2—figure supplement 3B*). We analysed these cells separately for both the wild-type siblings and *has/apkc* mutant embryos. Since there were only two 'short' cells in *has/apkc* mutants, we pooled them with medium heighted cells. We found that taller cells of *has/apkc* mutants had a smaller apical perimeter as compared to wild type embryos (*Figure 2—figure supplement 3 C1*; Mann Whitney U test, p<0.05). Interestingly, the apical perimeter of the medium/short heighted cells in *has/apkc* mutant was also smaller (*Figure 2—figure supplement 3 D1*; Mann Whitney U test, p<0.05). These data suggest that, irrespective of the cell height, mutant cells are more columnar than the sibling cells. In contrast, there was no significant effect on the morphology of the basal epidermal cells in *has/apkc* mutant embryos as compared to the siblings (*Figure 2 B2, B3*).

We further asked whether aPKC loss results in a perturbation of the polarised distribution of adherens junctions. We reasoned that systematic quantification would be required to reveal any effect of aPKC loss on polarity. To begin with, we identified the plane at which E-cadherin levels were the highest in each peridermal cell, and this level was considered to be 100%. Keeping this as a reference, we normalised the E-cadherin levels at other planes. The data were plotted as a scatter-plot, with the points binned in 15 bins. If the distribution remained unaltered, one would expect the maximum E-cadherin intensity to be achieved at the basal domain of peridermal cells. Thus, the cells having highest intensities at places other than the basal domain would contribute to the noise. We observed that, while sibling peridermal cells show a noisy but gradual increase in E-cadherin intensity along the apico-basal axis, the mutant peridermal cells exhibit increased noise and reduced robustness in polarised E-cadherin distribution (*Figure 2 A4*). Similar analysis on the basal epidermal cells

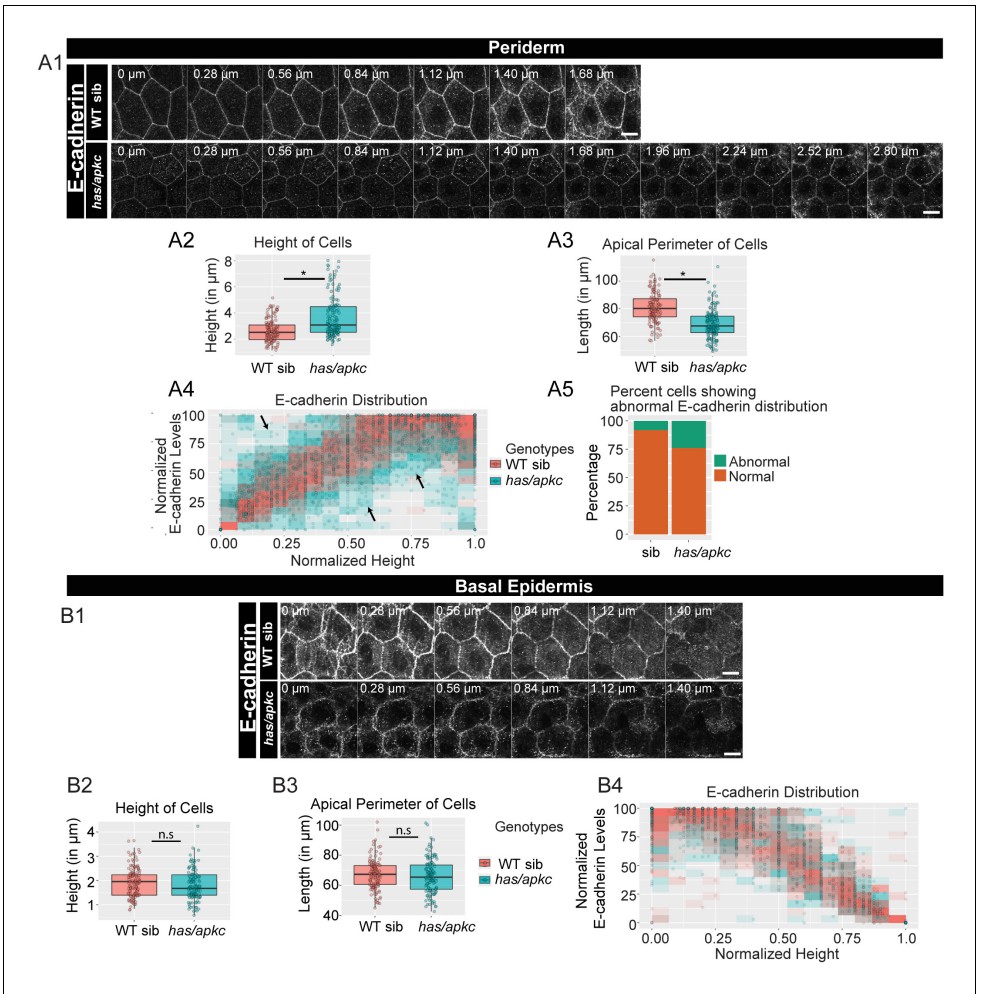

**Figure 2.** *has/apkc* function is required to restrict the noise in E-cadherin polarity. Immunolocalisation of E-cadherin along the apicobasal axis (0 μm is apical) in wild-type (WT) sibling and *has/apkc* mutant in the periderm (**A1**) and basal epidermis (**B1**) at 48hpf. Comparison of height of cells (**A2, B2**) and apical perimeter (**A3, B3**) in the periderm (**A2, A3**) and basal epidermis (**B2, B3**) of the two genetic conditions. Graphs showing polarised localisation of E-cadherin across normalised cell height in the periderm (**A4**) and basal epidermis (**B4**). A graph showing noise index in E-cadherin localisation in WT siblings versus *has/apkc* mutants (**A5**). Arrows in A4 pointing to the abnormal localisation of E-cadherin. Scale bars in A1, B1 are equivalent to 10 μm. AU = Arbitrary Units; WT = wild type, sib = sibling/s and *has/apkc* = *has/apkc* mutants. Asterisk indicates significant difference at p<0.05 and n.s. means non-significant difference observed by Mann Whitney U test. Source file with fluorescence intensities for periderm and basal epidermis is available as *Figure 2—source data 1* and *2*, respectively. Statistical analysis with p values for all the graphs of periderm and basal epidermis is available as *Figure 2—source data 3* and *4*, respectively.

The online version of this article includes the following source data and figure supplement(s) for figure 2:

**Source data 1.** Fluorescence intensities of E-cadherin and measurements of cellular attributes in the periderm of *has/apkc* mutants and WT siblings at 48 hpf.

**Source data 2.** Fluorescence intensities of E-cadherin and measurements of cellular attributes in the basal epidermis of *has/apkc* mutants and WT siblings at 48 hpf.

**Source data 3.** Statistical summaries of comparisons of cellular attributes and assessment of polarity in the periderm of *has/apkc* mutants and WT siblings presented in *Figure 2 A2-A5*.

**Source data 4.** Statistical summaries of comparisons of cellular attributes in the basal epidermis of *has/apkc* mutants and WT siblings presented in *Figure 2 B2-B3*.

**Figure supplement 1.** Localisation of aPKC in the periderm at different developmental stages.

**Figure supplement 2.** E-cadherin localisation with respect to Lyn-GFP in *has/apkc* mutants in the periderm.

**Figure supplement 3.** *has/apkc* regulates polarised localisation of E-cadherin independent of peridermal cell height.

*Figure 2 continued on next page*

*Figure 2 continued*

**Figure supplement 3—source data 1.** Fluorescence intensities of E-cadherin localisation and apical perimeter of tall cells of *has/apkc* mutants and WT siblings at 48 hpf for *Figure 2—figure supplement 3 C1-C3*.

**Figure supplement 3—source data 2.** Fluorescence intensities of E-cadherin localization and apical perimeter of medium/short cells of *has/apkc* mutants and WT siblings at 48 hpf for *Figure 2—figure supplement 3 D1-D3*.

**Figure supplement 3—source data 3.** Statistical summary of comparison of apical perimeter and polarity assessment between tall cells in *has/apkc* mutants and WT siblings presented in *Figure 2—figure supplement 3 C1-C3*.

**Figure supplement 3—source data 4.** Statistical summary of comparison of apical perimeter and polarity assessment between medium/short cells in has/apkc mutants and WT siblings presented in *Figure 2—figure supplement 3 D1-D3*.

**Figure supplement 4.** Demonstration of polarity defect in *has/apkc* mutant embryos in live expressing exogenous E-cadherin-mCherry.

**Figure supplement 4—source data 1.** Fluorescence intensities of E-cadherin-mCherry in the periderm of *has/apkc* mutants and WT siblings imaged live at 48hpf.

**Figure supplement 4—source data 2.** Statistical summaries of comparisons of cellular attributes and assessment of polarity in *has/apkc* mutants and WT siblings.

---

did not show any obvious effects on the polarised distribution of E-cadherin in the *has/apkc* mutant embryos (*Figure 2 B4*).

To estimate the 'noise index' in the periderm a linear regression was fitted to the sibling data shown in *Figure 2 A4* (*Figure 2—figure supplement 3A*; $R^2$ = 0.731). Next, we identified and counted the *has/apkc* mutant and sibling data points lying outside the 99% prediction interval for the siblings. Cells with even a single data point lying outside 99% interval were considered as having abnormal distribution of E-cadherin localisation. The noise index (abnormal cells*100/total cells) thus estimated was 24% for *has/apkc* mutant as compared to 8% for the siblings (*Figure 2 A5*).

Since *has/apkc* mutant periderm consists of cells that are taller than or comparable to wild type in height, we further asked whether this noise arises due to a variation in cell heights. We separately analysed the polarised distribution of E-cadherin in 'tall' and 'medium/short' cells, as classified earlier, in the periderm of *has/apkc* mutant embryos using the data presented in *Figure 2 A4*. Interestingly, in the periderm of *has/apkc* mutants, irrespective of the cell heights, a decrease in the robustness of polarisation and an increase in noise index was observed (*Figure 2—figure supplement 3, C3, D2, D3*). These data suggest that the effect on E-cadherin polarisation in *has/apkc* mutant embryos is independent of the cell heights.

We further performed the polarity analysis by analysing localisation of exogenously expressed E-cadherin-mCherry in live embryos. This analysis corroborated the findings that in the absence of *has/apkc* function, peridermal cells show a decrease in membrane localisation of E-cadherin-mCherry, reduction in the robustness of E-cadherin polarisation and increase in the noise index as compared to WT siblings (*Figure 2—figure supplement 4A,D,E*). Interestingly, injection of *e-cadherin-mCherry* mRNA, which presumably increases the total pool of E-cadherin (endogenous + exogenous), specifically rescued the cell-height phenotype in the periderm of *has/apkc* mutants over the control injections of *CAAX-mCherry* mRNA (*Figure 2—figure supplement 4B*, Kruskal Wallis test with Dunn's post hoc test; significance values in *Figure 2—figure supplement 4—source data 2*). This exogenously added E-cadherin failed to rescue the smaller apical perimeter of the mutant peridermal cells. (*Figure 2—figure supplement 4C*). These data confirmed that aPKC regulates the polarised distribution of E-cadherin in the periderm via the cell polarity pathway and not by controlling the cell height or levels of E-cadherin.

We then checked for the effects on adherens junctions in *pen/lgl2* background. Our preliminary data at 48hpf showed inconsistent result with respect to E-cadherin localisation presumably due to the presence of maternally derived Lgl2 at the earlier stages of development (*Raman et al., 2016*; *Sonawane et al., 2005*). We therefore restricted our analysis to a later developmental stage at 72hpf (*Figure 3 A1, B1*). We found a slight reduction in the E-cadherin intensities at various z-levels in both periderm and basal epidermis (*Figure 3 A1, B1*). We also observed a slight increase in the peridermal cell-height and in basal cell-perimeter in *pen/lgl2* mutant embryos (*Figure 3 A2, B3*; Mann Whitney U test, p<0.05). However, there was no effect on polarised distribution of E-cadherin in *pen/lgl2* mutants (*Figure 3 A4, B4*).

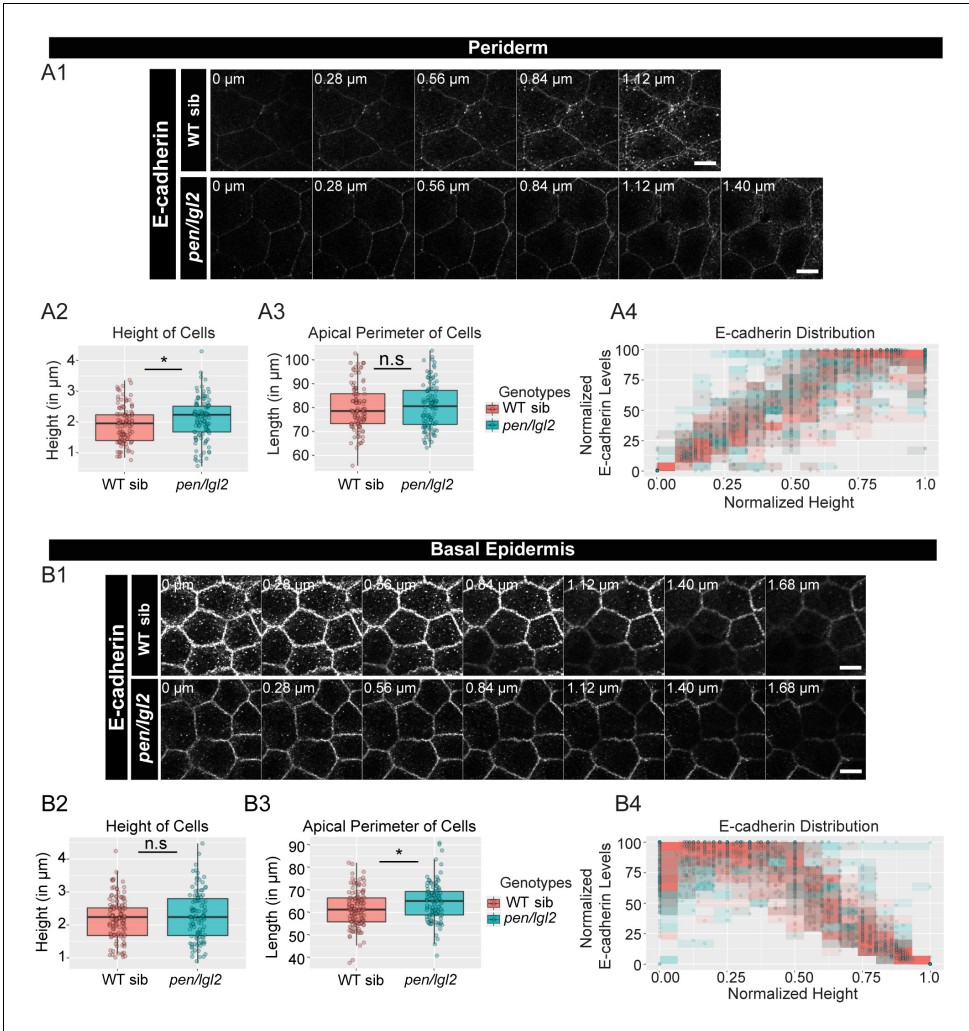

**Figure 3.** The loss of *pen/lgl2* function has no effect on E-cadherin polarity. Immunolocalisation of E-cadherin at various cell heights (0 μm is apical) along the apicobasal axis in wild-type sibling (WT sib) and *pen/lgl2* mutant in the periderm (**A1**) and basal epidermis (**B1**) at 72hpf. Comparison between WT siblings and *pen/lgl2* mutants in the periderm (**A2–A4**) and the basal epidermis (**B2–B4**) for height of cells (**A2, B2**) and apical perimeter (**A3, B3**). Graphs showing polarised distribution of E-cadherin across normalised cell height in the periderm (**A4**) and basal epidermis (**B4**) in WT siblings and *pen/lgl2* mutants. Scale bar corresponds to 10 μm in A1 and B1. AU = Arbitrary Units; WT = wild type and sib = sibling/s. Asterisk indicates significant difference (p<0.05) and n.s. means non-significant difference by Mann Whitney U statistical test. Source file with fluorescence intensities for periderm and basal epidermis is available as *Figure 3—source data 1* and *2*, respectively. Statistical analysis with p values for all the graphs of periderm and basal epidermis is available as *Figure 3—source data 3* and *4*, respectively.

The online version of this article includes the following source data for figure 3:

**Source data 1.** Fluorescence intensities of E-cadherin localization and measurements of cellular attributes in the periderm of *pen/lgl2* mutants and WT siblings at 72hpf.

**Source data 2.** Fluorescence intensities of E-cadherin localization and measurements of cellular attributes in the basal epidermis of *pen/lgl2* mutants and WT siblings at 72hpf.

**Source data 3.** Statistical summaries of comparisons of cellular attributes in the periderm of *pen/lgl2* mutants and WT siblings presented in *Figure 3 A2-A3*.

**Source data 4.** Statistical summaries of comparisons of cellular attributes in the basal epidermis of *pen/lgl2* mutants and WT siblings presented in *Figure 3 B2-B3*.

To conclude, *has/apkc* function is essential to maintain the robustness in adherens junction polarity along the apicobasal axis in the peridermal cells whereas *pen/lgl2* function seems dispensable in this process.

## E-cadherin in one epidermal layer controls the E-cadherin localisation in the juxtaposed epidermal layer in a layer non-autonomous manner

E-cadherin is one of the crucial factors involved in epidermal cell adhesion (*Tinkle et al., 2008*; *Tunggal et al., 2005*). In the zebrafish bilayered epidermis, E-cadherin localises to the basolateral domain of the peridermal cells and to the apico-lateral domain of the basal epidermal cells. Since it is involved in homophilic adhesion, it is likely that lateral membrane localisation indicates adhesion between two neighbouring peridermal cells and two basal epidermal cells (*Figure 1 A1, A2*; *Figure 1—figure supplements 1* and *2*). Similarly, the localisation of E-cadherin to the basal domain of the peridermal cells and apical domain of the juxtaposed basal epidermal cells reflects cadherin-mediated adhesion between the periderm and basal epidermal cells.

We hypothesised that, due to homophilic adhesion, localisation of E-cadherin in one layer would determine the membrane levels in the other layer. To test this hypothesis, we knocked down *e-cadherin* in a clonal manner, either in the periderm or the basal epidermis, and checked the E-cadherin levels at the membrane in the other layer. To achieve this, we injected *e-cadherin* morpholino along with GFP mRNA, as a tracer, in one of the blastomeres at 16–64 cell stage. To prove that the morpholinos injected this way partition together with the mRNA, we injected 3'- Carboxyfluorescein tagged standard control morpholino with mCherry mRNA. We found that mCherry labelled peridermal or basal epidermal clones showed tight association with the fluorescent morpholino (*Figure 4—figure supplement 1A,B*). Of the 23 clones analysed this way, 21 showed co-localisation of mCherry with the morpholino. None of the clones showed inter layer diffusion and 2/23 clones either did not show co-localisation or showed diffusion of morpholino beyond mCherry expression within the layer. We knocked down *e-cadherin* function this way in either the periderm or the basal epidermis. We observed that the reduction of E-cadherin levels was not restricted to the peridermal clones, which received the morpholino, but was also seen in the underlying basal epidermal cells. Of the 23 peridermal clones analysed, 19 clones showed a layer non-autonomous effect (*Figure 4A*). This was reciprocally true for the basal epidermal clones. Of 44 basal epidermal clones analysed, 36 showed layer non-autonomous effect on membrane levels of E-cadherin in the periderm (*Figure 4D*). Most importantly, this reduction in the levels of E-cadherin was not limited to the interface between the clone and underlying or overlying cells but was also observed at the lateral domains of the cells of the juxtaposed layer (*Figure 4A,D*). We further quantified E-cadherin levels in the morphant clones that showed qualitative effect by dividing the cell-boundaries into six categories as described ahead (See methods for details; *Figure 4—figure supplement 1 C1, C2*). The quantifications from three independent experiments were analysed separately (*Figure 4B,C,E,F*; *Figure 4—figure supplement 2 A1-D2*). This analysis confirmed that, whether in the periderm or basal epidermis, the clonal knockdown of *e-cadherin* resulted in a marked and statistically robust reduction in E-cadherin levels within the clone (CC) and at the cell-boundaries in the juxtaposed layer that completely overlapped (CO) with the clone. The effect on the non clone-clone boundaries (NC) was moderate. These quantifications also revealed a reduction in E-cadherin at the boundaries between the nearby non-clonal cells (NN) and the boundaries in the juxtaposed layer that partially overlap (PO) or do not overlap (NO) with the morphant clones. However, this effect on PO, NO and NN was statistically not as robust and varied across the three sets (*Figure 4B,C,E,F*; *Figure 4—figure supplement 2 A1-D2*; Kruskal Wallis test with Dunn's post hoc test; significance values in *Figure 4—source data 5*, *Figure 4—figure supplement 2—source data 9*). It is also important to note that the layer non-autonomous effect on the juxtaposed layer was stronger near the interface and decreased towards apical (in periderm) or basal (in basal epidermis).

As shown previously (*Kane et al., 2005*; *Shimizu et al., 2005*), the whole body knock down of *e-cadherin* function is lethal and the embryo does not develop past the epiboly stages. Co-injecting *e-cadherin* mRNA along with morpholino rescued this early phenotype, proving the specificity of the morpholino used in this study (*Figure 4—figure supplement 3*).

We conclude that E-cadherin in one epidermal layer controls the E-cadherin localisation in the cells of the juxtaposed layer, not only at the interface of the two layers but also at the lateral domain of the cells, in a layer non-autonomous manner.

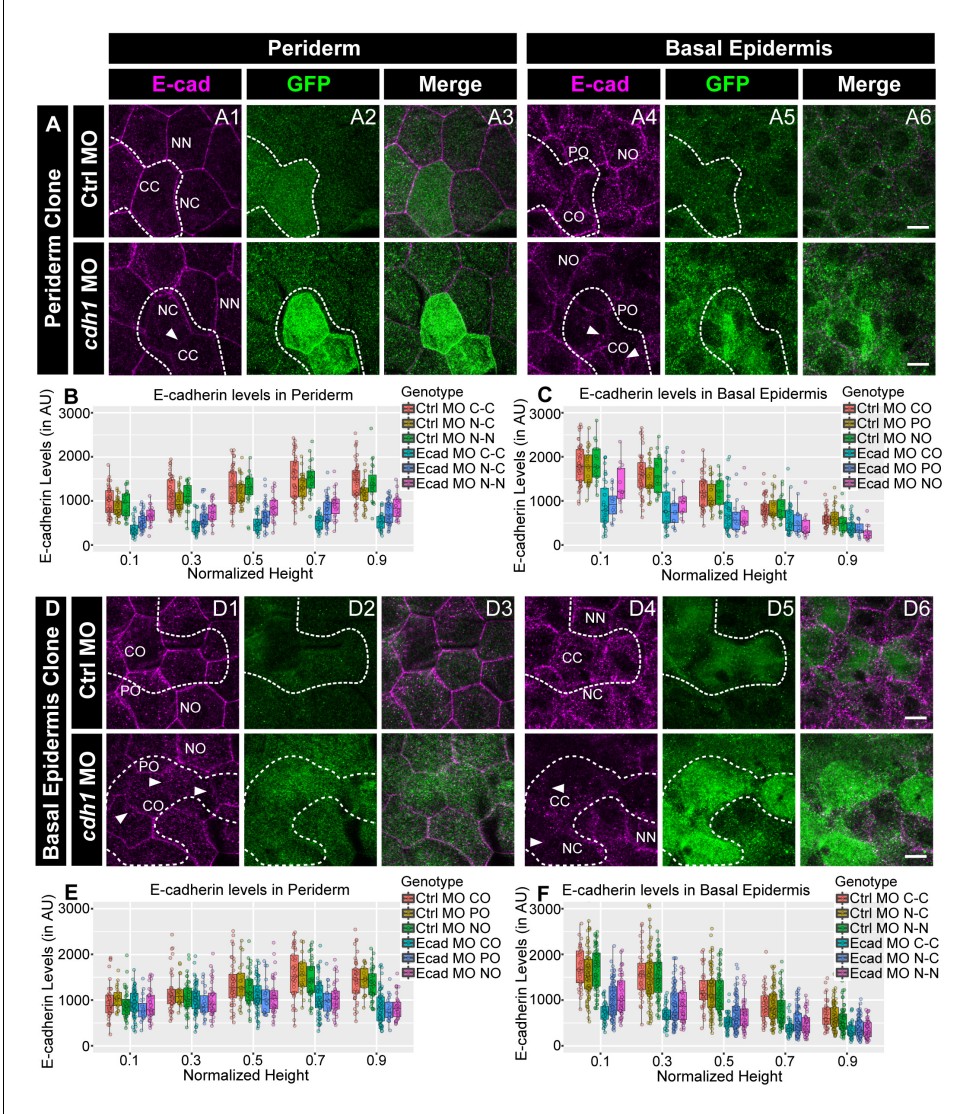

**Figure 4.** E-cadherin in one epidermal layer regulates E-cadherin localisation in the adjacent layer in a layer non-autonomous manner. Confocal images of E-cadherin (magenta) localisation in GFP clones (green) carrying *cadherin-1* morpholino (*cdh-1* MO) and control morpholino (Ctrl MO) in the periderm (**A**) and basal epidermis (**D**). (**A1–A3**) shows peridermal clone and its effect on E-cadherin localisation in the basal epidermal cells (**A4–A6**). (**D4–D6**) show a basal epidermis clone and its effect on E-cadherin localisation in the periderm (**D1–D3**). Graphs showing effect on E-cadherin levels across normalised cell height in the periderm (**B, E**) and basal epidermis (**C, F**) across different boundaries upon knockdown of *e-cadherin* in a clonal manner. The boundaries Clone-Clone (C–C), Clone-Non-clone (N–C), and Non clone-Non clone (N–N) are within the same layer with respect to the clones whereas boundaries showing Complete Overlap (CO), Partial Overlap (PO), and No Overlap (NO) are in the cells of the juxtaposed layer. Dotted line in (**A1, A2**) represents the position of the peridermal clone and the basal epidermal region below the clone (**A4, A5**). Similarly, dotted lines in (**D4, D5**) mark the clones in the basal epidermis and the peridermal region above the clone (**D1, D2**). Arrowheads point to the loss/reduction in E-cadherin staining at the cell membranes. Scale bar represents 10 μm in (**A6, D6**). AU = Arbitrary Units. Source file with fluorescence intensities for peridermal clone and E-cadherin analysis at the boundaries in the periderm (**B**) and basal epidermis (**C**) is available as *Figure 4—source data 1* and *2*, respectively. Fluorescence intensities for basal epidermal clones and analysis at the boundaries in periderm (**E**) and basal epidermis (**F**) is available as *Figure 4—source data 3* and *4*, respectively. Statistical analysis consisting of intensity comparisons is available as *Figure 4—source data 5*.

The online version of this article includes the following source data and figure supplement(s) for figure 4:

*Figure 4 continued on next page*

*Figure 4 continued*

**Source data 1.** Fluorescence intensities of E-cadherin localisation along different boundaries of the peridermal cells when the clones are in the periderm for *Figure 4B*.

**Source data 2.** Fluorescence intensities of E-cadherin localisation along different boundaries of the basal epidermal cells when clones are in the periderm for *Figure 4C*.

**Source data 3.** Fluorescence intensities of E-cadherin localisation along different boundaries of the peridermal cells when clones are in the basal epidermis for *Figure 4E*.

**Source data 4.** Fluorescence intensities of E-cadherin localisation along different boundaries of the basal epidermal cells when clones are in the basal epidermis for *Figure 4F*.

**Source data 5.** Statistical summaries of comparisons between E-cadherin levels along *cdh1* MO and Ctrl MO boundaries presented in *Figure 4B, C, E, F*.

**Figure supplement 1.** Analysis of co-segregation of 3'- Carboxyfluorescein Standard Morpholino and mCherry; schematic showing classification of boundaries used for E-cadherin quantifications.

**Figure supplement 2.** Quantification of E-cadherin levels in two additional experimental sets to assess the cell autonomous and non-autonomous effect of e-*cadherin* knockdown.

**Figure supplement 2—source data 1.** Fluorescence intensities of E-cadherin localisation along different boundaries of the peridermal cells when clones are in the periderm for *Figure 4—figure supplement 2 A1* (second set).

**Figure supplement 2—source data 2.** Fluorescence intensities of E-cadherin localisation along different boundaries in the basal epidermis when clones are in the periderm for *Figure 4—figure supplement 2 A2* (second set).

**Figure supplement 2—source data 3.** Fluorescence intensities of E-cadherin localisation along different boundaries in the periderm when clones are in the basal epidermis for *Figure 4—figure supplement 2 B1* (second set).

**Figure supplement 2—source data 4.** Fluorescence intensities of E-cadherin localisation along different boundaries in the basal epidermis when clones are in basal epidermis for *Figure 4—figure supplement 2 B2* (second set).

**Figure supplement 2—source data 5.** Fluorescence intensities of E-cadherin localisation along various boundaries in the periderm when clones are in the periderm for *Figure 4—figure supplement 2 C1* (third set).

**Figure supplement 2—source data 6.** Fluorescence intensities of E-cadherin localisation along different boundaries in the basal epidermis when clones are in the periderm for *Figure 4—figure supplement 2 C2* (third set).

**Figure supplement 2—source data 7.** Fluorescence intensities of E-cadherin across different boundaries in the periderm when clones are in the basal epidermis for *Figure 4—figure supplement 2 D1* (third set).

**Figure supplement 2—source data 8.** Fluorescence intensities of E-cadherin across different boundaries in the basal epidermis when the clones are in the basal epidermis for *Figure 4—figure supplement 2 D2* (third set).

**Figure supplement 2—source data 9.** Statistical summaries of comparisons between E-cadherin levels along *cdh1* MO and Ctrl MO boundaries presented in *Figure 4—figure supplement 2 A1, A2, B1, B2, C1, C2, D1, D2*.

**Figure supplement 3.** Rescue of e-*cadherin* (*cdh-1*) morphant phenotype by e-*cadherin* mRNA.

## E-cadherin regulates localisation of aPKC and Lgl

The apicobasal polarity in simple epithelia such as follicular epithelium or blastoderm epithelium in *Drosophila*, is regulated by adherens junctions (*Müller and Wieschaus, 1996*; *Tanentzapf et al., 2000*). We asked whether E-cadherin regulates polarity in the zebrafish bilayered embryonic epidermis. To answer this question, we resorted to clonal analysis using *e-cadherin* morpholino. To ensure that the knockdown of *e-cadherin* leads to loss of adherens junctions, we analysed localisation of β-catenin. Our analysis revealed that reduction in E-cadherin levels led to a decrease in membrane associated β-catenin, indicating a significant reduction in adherens junction formation (*Figure 5—figure supplement 1*).

We further analysed the effect of *e-cadherin* knockdown on aPKC and Lgl2 localisation. In the morphant peridermal clones, aPKC was present basolaterally in the sub-cortical region, in addition to its apical localisation (*Figure 5A*). Out of 18 clones analysed, 14 showed this effect. Our results from the previous section indicate that the knockdown of *e-cadherin* in the basal epidermis results in a reduction of peridermal E-cadherin in a layer non-autonomous manner. Therefore, we checked aPKC localisation in the periderm overlying a basal epidermal morphant clones. Our analysis did not reveal any effect on aPKC localisation in the periderm when *e-cadherin* was knocked down in basal epidermal cells (*Figure 5—figure supplement 2*).

The localisation of aPKC to ectopic cytoplasmic sites in the absence of *e-cadherin* function in the peridermal cells raised a possibility that localisation of other apical proteins is also perturbed. It has been shown that the formation of tight junctions in the mammalian epidermis requires E-cadherin

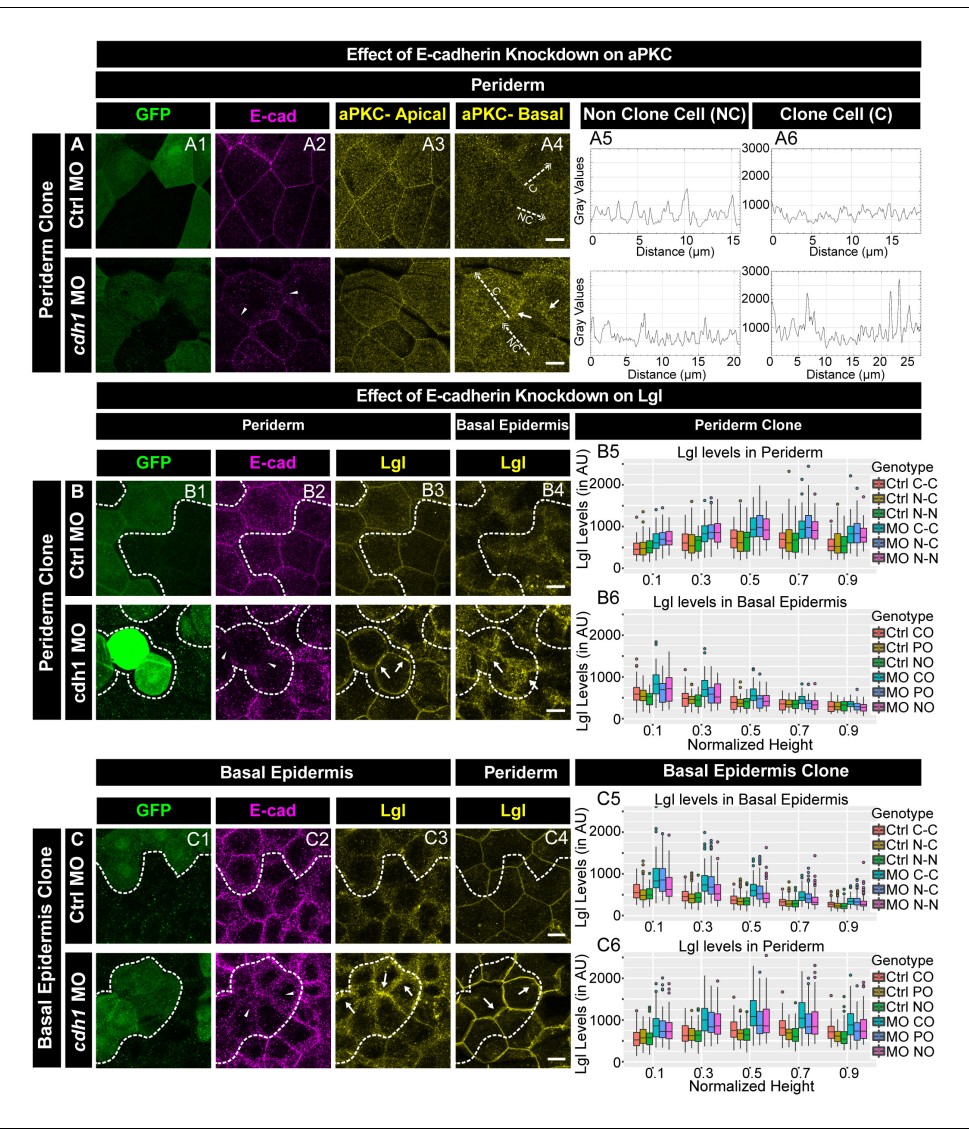

**Figure 5.** E-cadherin regulates localisation of aPKC and Lgl2 in the epidermis. Confocal images showing GFP marked clones (**A1, B1, C1**) marking *cadherin-1* morpholino (*cdh-1* MO) or control MO (Ctrl MO) and immunolocalisation of E-cadherin (**A2, B2, C2**), aPKC (**A3, A4**) and Lgl2 (**B3, B4, C3, C4**). Arrowhead marks the loss of E-cadherin in morphant cells (**A2, B2, C2**). Different z-stacks depicting apical (**A3**) or basal domains (**A4**) of peridermal cells showing aPKC immunostaining. Arrow shows enrichment of aPKC in the basal domain of the *cdh-1* morphant cells (**A4**). Intensity profiles for aPKC in the basal domain in non-clone (NC) cells (**A5**) and clone (C) cells (**A6**) were measured along the dotted arrows (in **A4**). Lgl staining in the periderm (**B3, C4**) or basal epidermis (**B4, C3**). Arrows show the high levels of Lgl in the morphant or juxtaposed cells (**B3, B4, C3, and C4**) upon *e-cadherin* knockdown. Graphs showing levels of Lgl across normalised cell height in the periderm (**B5, C6**) and basal epidermis (**B6, C5**) across different boundaries mentioned. Dotted line indicates the position of the clone in the periderm (**B1–B3**) or basal epidermis (**C1–C3**) and corresponding region in basal epidermis (**B4**) or periderm (**C4**), respectively. Scale bar represents 10 µm in (**A4, B4, C4**). AU = Arbitrary Units. Source file with fluorescence intensities for peridermal clone and boundary analysis in periderm (**B5**) and basal epidermis (**B6**) is available as *Figure 5—source data 1* and *2*, respectively. Fluorescence intensities for basal epidermis clone and analysis of boundaries in basal epidermis (**C5**) and periderm (**C6**) is available as *Figure 5—source data 3* and *4*, respectively. Statistical analysis of intensity comparisons along with p values is available as *Figure 5—source data 5*. The online version of this article includes the following source data and figure supplement(s) for figure 5:

**Source data 1.** Fluorescence intensities of Lgl along different boundaries in the periderm when clones are in the periderm for *Figure 5 B5*.

*Figure 5 continued on next page*

*Figure 5 continued*

**Source data 2.** Fluorescence intensities of Lgl along different boundaries in the basal epidermis when clones are in the periderm for *Figure 5 B6*.

**Source data 3.** Fluorescence intensities of Lgl along different boundaries in the basal epidermis when the clones are in the basal epidermis for *Figure 5 C5*.

**Source data 4.** Fluorescence intensities of Lgl along different boundaries in periderm when the clones are in basal epidermis for *Figure 5 C6*.

**Source data 5.** Statistical summaries of comparisons of Lgl levels between *cdh1* MO and Ctrl MO boundaries presented in *Figure 5 B5, B6, C5, C6*.

**Figure supplement 1.** *e-cadherin* knockdown results in the loss of β-catenin localisation.

**Figure supplement 2.** *e-cadherin* knockdown in the basal epidermis does not have any effect on aPKC localisation in the periderm or basal epidermis.

**Figure supplement 3.** *e-cadherin* knockdown has no effect on localisation of ZO-1.

**Figure supplement 4.** EC1-EC2 domain of E-cadherin is required to maintain Lgl2 levels in the epidermis.

function (*Tinkle et al., 2008*; *Tunggal et al., 2005*). Hence, we analysed localisation of ZO-1, a tight junction marker, in the periderm upon *e-cadherin* knockdown. We did not observe mislocalisation of ZO-1 (*Figure 5—figure supplement 3*), indicating that the loss of E-cadherin does not affect the apical domain in general but has a specific effect on aPKC.

To probe the effect of *e-cadherin* knockdown on basolateral polarity components, we analysed localisation of Lgl2. The knockdown of *e-cadherin* in either the peridermal or the basal epidermal clones showed an increase in Lgl2 levels (*Figure 5B,C*). In addition, E-cadherin deficiency in the periderm caused an increase in Lgl at the lateral membrane of the underlying basal epidermal cells (*Figure 5B*). Similarly, E-cadherin deficiency in the basal epidermis caused an increase in Lgl2 levels at the lateral domain of the juxtaposed peridermal cells (*Figure 5C*). This non-autonomous effect was observed in 27 clones out of total 34 clones. The layer autonomous as well as non-autonomous increase in Lgl2 levels in the absence *e-cadherin* function was patchy. Further quantification of the Lgl levels in clones that showed qualitative effect, confirmed a robust increase in Lgl levels in a layer autonomous as well as layer non autonomous manner. (*Figure 5 B5, B6, C5, C6*; Kruskal Wallis test with Dunn's post hoc test, significance values in *Figure 5—source data 5*).

To understand whether the core function of E-cadherin in cell adhesion or its interaction with cytoskeleton is responsible for regulation of polarity, we synthesised two GFP tagged deletion constructs of E-cadherin. In the first construct, the first two extracellular domains (ΔEC1-EC2-Ecad), which would abrogate homophilic interactions between cadherin molecules (*Kardash et al., 2010*; *Leckband and de Rooij, 2014*), were deleted. In the second construct, the β-catenin binding domain - which is important for cytoskeletal interaction- was deleted (*Gottardi et al., 2001*; *Leckband and de Rooij, 2014*; *Niessen et al., 2011*). Unlike full-length construct, the mRNA synthesised from the deletion constructs and injected in a clonal manner did not yield any localisation of these deletion forms of E-cadherin. This precluded the possibility of complementing the morpholino induced E-cadherin deficiency by these two E-cadherin deletions. However, upon plasmid injection, ΔEC1-EC2-Ecad localised to the membrane (*Figure 5—figure supplement 4 A1, B1, C1*). While the total membrane-bound E-cadherin pool seem to increase in these clones, presumably due to presence of endogenous E-cadherin and exogenously expressed mutant version, we did not observe any non-autonomous effect on E-cadherin localisation in cells of the juxtaposed layer (*Figure 5—figure supplement 4 A2, A4, B2, B4*). Interestingly, of the 19 clones expressing ΔEC1-EC2-Ecad version, 16 showed increased levels of Lgl2 in a layer autonomous as well as non-autonomous manner (*Figure 5—figure supplement 4 A3, A5, B3, B5*). However, there was no detectable change in aPKC levels or localisation in 15 such clones analysed (*Figure 5—figure supplement 4 C3, C4*). Despite repeated attempts, we did not observe membrane localisation of plasmid encoded mutant form of E-cadherin in which β-catenin binding domain was deleted. This could possibly be due to the increased endocytosis of the mutant version of the E-cadherin (*Chen et al., 1999*; *Hall et al., 2019*).

To conclude, the loss of *e-cadherin* function results in a loss of β-catenin localisation, suggesting a reduction in adherens junction formation. E-cadherin restricts the localisation of aPKC to the apical domain in the periderm in a cell autonomous manner, and controls the localisation of Lgl in a layer

autonomous and non-autonomous manner. Our results suggest that extracellular domains of E-cadherin play important role in controlling localisation of Lgl2 within and across the layers.

## Discussion

The developing epidermis, developing urinary bladder and epithelia in mammary ducts and sweat gland ducts are bilayered in nature whereas the adult epidermis, oesophagus lining and cornea are multi-layered epithelial systems (*Cui and Schlessinger, 2015*; *Kierszenbaum and Tres, 2015*; *Newman and Antonakopoulos, 1989*; *Standring, 2016*). How polarity is regulated in such bi-layered and multilayered epithelia has remained poorly understood. Here we have used zebrafish developing epidermis, a bilayered epithelium, as a model to show that adherens junctions are localised in a polarised manner. In addition, we have also unravelled the importance of aPKC and E-cadherin in the maintenance of cell-polarity in epidermal layers.

In vertebrate as well as invertebrate simple epithelia, zonula adherens forms at the apicolateral domain (*Grawe et al., 1996*; *Kierszenbaum and Tres, 2015*; *Liu et al., 2007*; *Standring, 2016*; *Wodarz et al., 2000*). Besides the apical belt of zonula adherens, adherens junctions are also reported to be present in the lateral domain in MDCK cells and follicular epithelium in *Drosophila* (*Tanentzapf et al., 2000*; *Zinkl et al., 1996*). Though shown to be present at the lateral domains, it remains unclear how adherens junctions are distributed in multi-layered epithelial systems (*Hirai et al., 1989*; *Horiguchi et al., 1994*; *Ishiko et al., 2003*; *Tunggal et al., 2005*). Here we have demonstrated that adherens junctions are distributed in a polarised manner in the zebrafish bilayered embryonic epidermis. Peridermal cells show an increase in the level of E-cadherin and β-catenin, and thus adherens junctions, from the apical to the basal domain whereas the basal epidermal cells show a reverse trend. Thus, the interface of the two layers has the maximum E-cadherin levels. This is a unique system in which the two juxtaposed layers of the epithelium show reversed polarity with respect to the localisation of adherens junctions. Consistent with this, we did not observe localisation of aPKC in the apical domain of the basal epidermal cells during early development. It appears that the molecular identity of the histological apical domain of the basal epidermal cells remains undefined at the early stages of the epidermis development. Alternatively, it may be a feature of the bilayered epithelial systems in general. The basal epidermal cells may get the apical identity at the 'molecular' level possibly at a later stage when the polarity of the basal cells becomes important for the asymmetric cell division during stratification, as shown in mice (*Helfrich et al., 2007*; *Niessen et al., 2013*; *Osada et al., 2015*).

The polarity proteins regulate the formation and localisation of adherens junctions (*Bossinger et al., 2001*; *Qin et al., 2005*; *Rodriguez-Boulan and Macara, 2014*; *Suzuki et al., 2002*; *Yamanaka et al., 2006*). *apkc* function has been shown to be important for the formation and/or localisation of adherens junction in *Drosophila* neuroblast as well as in vertebrate epithelia including neuroepithelium (*Horne-Badovinac et al., 2001*; *Imai et al., 2006*; *Kishikawa et al., 2008*; *Osada et al., 2015*; *Rolls et al., 2003*; *Suzuki et al., 2002*). Our data indicate that aPKCλ/ι function is required to reduce the noise in E-cadherin localisation to generate robust polarised distribution of adherens junctions. A systematic quantification followed by rigorous data analysis is required to unravel this phenotype. Importantly, the decreased robustness in polarity is not a consequence of changes in the peridermal cell morphologies or E-cadherin levels, suggesting that aPKC controls the polarised E-cadherin localisation primarily via the polarity pathway. Intriguingly, the basolateral regulator *lgl2* does not seem to be required for the polarised distribution of E-cadherin. However, this could be due to the redundant functioning of the two *lgl* paralogues or due to maternally derived Lgl2, which remains to be tested.

E-cadherin participates in homophilic adhesion (*Ahrens et al., 2002*; *Nagar et al., 1996*; *Tomschy et al., 1996*). Therefore, consistent with previous studies (*Shamir et al., 2014*; *Vleminckx et al., 1991*; *Yokoyama et al., 2001*), we observed a reduction in E-cadherin localisation at the boundary between cells that are wild type and cells that are deficient for E-cadherin. Furthermore, our analyses presented here indicate that the knockdown of *e-cadherin* in a clonal manner in one layer results in the loss of E-cadherin localisation at the interface between the two layers. However, interestingly, the localisation of E-cadherin at the lateral domains is also affected in cells in the layer abutting the E-cadherin deficient clone. This clearly indicates that E-cadherin in one layer controls the localisation of E-cadherin at the lateral domains of the juxtaposed cells in a layer non-

autonomous manner. To the best of our knowledge, this is the first study to show a layer non-autonomous role of E-cadherin in a multi-layered epithelium. Our data, despite the variations across experiments, also indicate that weaker effect of *e-cadherin* loss may spread beyond the clone and the cells of the juxtaposed layer that are directly in contact with the clone. More sensitive methods of E-cadherin detection levels are required to reproducibly detect such weaker effect away from the deficient clone.

Earlier studies have shown that the disruption of adherens junctions leads to a loss of polarity and epithelial integrity in simple epithelia such as *Drosophila* blastoderm and follicular epithelium (*Müller and Wieschaus, 1996*; *Tanentzapf et al., 2000*). In multi-layered epithelia, a loss of *e-cadherin* function leads to mislocalisation of aPKC and improper formation of tight junctions (*Rübsam et al., 2017*; *Tunggal et al., 2005*). We observed that *e-cadherin* loss leads to mislocalisation of aPKC in the periderm. In addition to the tight apical localisation, increased non-membranous accumulation of aPKC is seen in the basal part of the E-cadherin deficient peridermal cells. This aPKC phenotype is not apparent in the peridermal cells when the *e-cadherin* is knocked down in the basal epidermis resulting in decrease in E-cadherin levels in the periderm. It is plausible that the weaker reduction in E-cadherin levels in the periderm upon *e-cadherin* knockdown in the basal epidermis may not be enough to mislocalise aPKC. The effect on the apical domain is limited only to aPKC. We did not observe any effect on localisation of tight junction component ZO-1. Presumably, the presence of maternally deposited E-cadherin protein facilitates the tight junction formation in EVL cells. The effect on polarity, upon *e-cadherin* knockdown, is also manifested by increased localisation of Lgl. The depletion of E-cadherin results in a layer autonomous as well as layer non-autonomous increase in Lgl localisation. Our Lgl data reveal that the polarity of one layer is linked with the polarity of the neighbouring layer via E-cadherin.

How is the E-cadherin localisation at the lateral domain in one layer achieved by E-cadherin present in another layer? How does E-cadherin regulate Lgl levels? How does absence of E-cadherin in clones propagated to the cell-boundaries that are away from the clones? E-cadherin regulates homophilic cell adhesion, achieves mechanical coupling, participates in cell signalling and controls cortical tension in epithelial cells (*Borghi et al., 2012*; *Chappuis-Flament et al., 2001*; *Gottardi et al., 2001*; *Koch et al., 1999*; *Lecuit and Yap, 2015*; *Maître et al., 2012*; *Martin et al., 2009*; *Onder et al., 2008*; *Takeichi, 1991*; *Tunggal et al., 2005*). It is likely that the layer autonomous and layer non-autonomous regulation of adherens junction and Lgl2 localisation happens via some of these mechanisms. The fact that expression of EC1-EC2 domain deleted form of E-cadherin does not influence the E-cadherin localisation layer-non-autonomously but alters Lgl localisation in autonomous as well as non-autonomous manner, indicates that homophilic interaction dependent mechanism(s) may not play a role in regulation of Lgl2 localisation. It is therefore likely that additional tension mediated mechanism(s) (*Borghi et al., 2012*; *Conway et al., 2017*; *Rübsam et al., 2017*; *Simões et al., 2010*) might be contributing towards this regulation. The effect on Lgl2 and E-cadherin localisation away from E-cadherin deficient clonal boundaries can be explained via such tension-based mechanisms. Clearly, more analysis is required to understand the mechanism of layer non-autonomous control of adherens junction formation and layer autonomous as well as layer non-autonomous regulation of Lgl2 localisation by E-cadherin.

Our data presented here along with the previous data (*Sonawane et al., 2005*; *Sonawane et al., 2009*) allow us to propose a model for polarisation of the bilayered epidermis (*Figure 6*). In zebrafish, the periderm differentiates from the enveloping layer (EVL). The basal epidermis differentiating from the ventral non-neural ectoderm (*Bakkers et al., 2002*; *Chang and Hwang, 2011*; *Kimmel et al., 1990*) underlies the periderm. EVL cells are polarised along the apicobasal axis (*Fukazawa et al., 2010*; *Lepage et al., 2014*) and are likely to pass on this polarity to the peridermal cells. As the first step in our model (*Figure 6*), aPKC regulates the polarised localisation of E-cadherin in the periderm. The basal epidermal cells do not exhibit apical localisation of aPKC (*Figure 2—figure supplement 1*), consistently aPKC does not control polarity in the basal epidermal cells. During the early phase of epidermis development, E-cadherin begins to localise in the basal epidermal cells at 22 hpf (Arora and Sonawane, unpublished observations). The peridermal E-cadherin- via homophilic interactions - organises E-cadherin in the basal epidermis. We propose that, as a consequence of homophilic interactions, the higher levels of E-cadherin in the basal domain of the periderm allow greater localisation of E-cadherin in the apical domain of the juxtaposed basal cells. Thus, initiating polarity in the basal epidermis. In addition, E-cadherin controls the levels of Lgl2 at

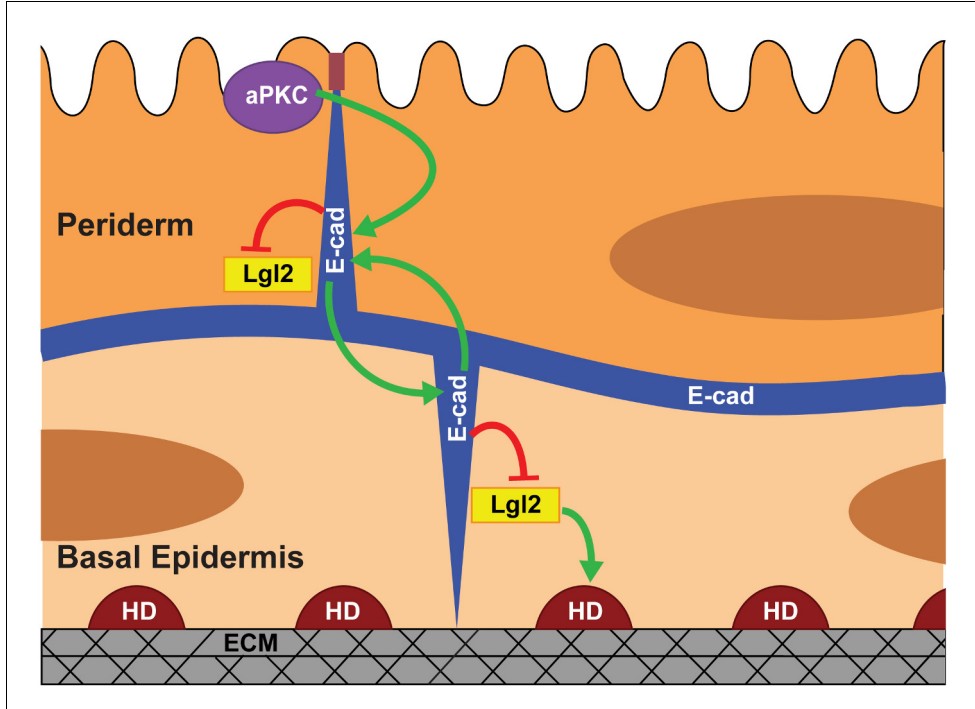

**Figure 6.** Model proposing stepwise polarisation of the zebrafish embryonic epidermis. The periderm, polarised at EVL stages, has apically localised aPKC. aPKC controls the polarisation of E-cadherin (E-cad). E-cadherin in one layer controls the localisation of E-cadherin in the other layer and negatively controls the Lgl2 levels at the cortex. Lgl2 further regulates formation of hemidesmosomes- junctions mediating cell-matrix adhesion in the basal epidermis. The polarity is setup in a stepwise manner wherein E-cadherin acts a transducer of polarity between the two layers. The number of cells/boundaries quantified for each figure is compiled as an excel sheet and is available as *Figure 6—source data 1*.

The online version of this article includes the following source data for figure 6:

**Source data 1.** Details of the numbers of cells, boundaries or clones analysed for each experiment.

the cortex (current analysis), which is essential for the formation hemidesmosomes that mediate basal cells-to-matrix adhesion (*Sonawane et al., 2005*; *Sonawane et al., 2009*). Whether Lgl in the basal epidermis is organised by peridermal E-cadherin or E-cadherin in the basal epidermis, remains unclear. Our results presented here also explain how E-cadherin and Lgl2 function antagonistically to regulate the hemidesmosome formation (*Sonawane et al., 2009*).

This model predicts that aPKC would regulate E-cadherin polarity in the basal epidermis via peridermal E-cadherin. Our current analysis doesn't support this prediction because in the *has/apkc* mutant E-cadherin polarity of the basal epidermal cells remains unaltered. However, it is important to note that in the absence of aPKC function, the E-cadherin polarity is noisy but not completely lost, possibly due to existence of redundant mechanisms. Such a partially polarised periderm may still be able to transduce the polarity cues to the basal epidermis to set up the E-cadherin polarity. Therefore, the possibility of aPKC regulating polarity in the basal epidermis via peridermal E-cadherin needs to be investigated further by eliminating the redundant mechanisms that control E-cadherin polarity in the periderm. Our proposed model and the approach establish a conceptual and analytical framework to systematically tease apart the mechanisms involved in polarity establishment across bilayered and multilayered epithelial systems.

# Materials and methods

## Fish strains

All experiments were done in Tübingen (Tü) strain unless mentioned otherwise. To mark the membrane the *Tg(cldb::lyn-EGFP)* fish (*Haas and Gilmour, 2006*) were used. Mutant strains with the following alleles were used, *heart and soul (has/apkc)^{m567}* (*Horne-Badovinac et al., 2001*), *penner (pen/lgl2)^{t06}* (*Sonawane et al., 2005*). The zebrafish maintenance and experimental protocols used in this study were approved by the institutional animal ethics committee.

## Microinjections

For microinjections, World Precision Instruments PV830 Pneumatic PicoPump was used. Embryos were injected at 1 cell stage or at 16–64 cell stage with 1 nl to 3 nl of 50 µM of *cdh1* morpholino (5'-ATCCCACAGTTGTTACACAAGCCAT-3') or its five base mismatch control morpholino (5'- A TCCGACACTTCTTACAGAACCCAT −3') or 50 µM of standard control morpholino tagged with 3'-Carboxyfluorescein (5'- CCTCTTACCTCAGTTACAATTTATA −3') or 200 ng/µl of mCherry tagged *e-cadherin* mRNA. For the clonal analysis, one cell at 16–64 cell stage was injected with the morpholino along with the GFP/mCherry RNA as a tracer. The zebrafish *e-cadherin-mCherry* construct (codon modified) was a gift from Prof. Erez Raz (Universität Münster).

The mRNA for GFP and *e-cadherin-mCherry*, were synthesised using mMESSAGE mMACHINE SP6 Transcription Kit (ThermoFisher – Invitrogen, AM1340) and purified using Micro Bio-Spin P-30 Gel Columns, Tris Buffer (RNase-free) (BIORAD, 732–6250).

The deletion constructs were synthesised in pMK-T and subcloned in pCS2+8-mCherry plasmid by GeneArt (ThermoFisher) based on the codon-modified version of *e-cadherin-mcherry*. The ΔEC1-EC2-Ecad construct lacked the extracellular domains EC1 and EC2 (K51-V223) (*Kardash et al., 2010*). The β-catenin deletion construct lacked the cytoplasmic C-terminal tail (G821-D864) (as based on *Gottardi et al., 2001*). Since neither of the two deletion constructs with the mCherry tag localised at the membrane, we subcloned the two deletion constructs along with the full length E-cadherin to pCS2+8-GFP. The subcloning was performed either by restriction digestion (BamHI-ClaI) or PCR based amplification. Clones were confirmed by sequencing. For injections, the plasmids were isolated using GeneJET Plasmid miniprep kit (ThermoFisher - K0503) and eluted in nuclease free water. ΔEC1-EC2-Ecad-GFP and Ecad-FL- GFP were injected at 70 ng/µl or 50 ng/µl concentrations, respectively.

## Immunostaining

For the whole mount immunostaining, embryos were fixed at the desired time points in 4% PFA in PBS, incubated at room temperature for 30 min, and then transferred to 4°C overnight. The embryos were then serially upgraded (30%, 50%, 70% Methanol in PBS) to 100% methanol and stored overnight at −20°C. The embryos were then serially downgraded to PBS and given five washes of PBST (0.8% Triton-X 100 in PBS). The embryos were blocked in 10% NGS (Jackson Immuno Research Labs, 005-000-121) in PBST for four hours. The embryos were then incubated overnight at 4°C on a rotor in primary antibodies in PBST with 1% NGS using following dilutions: E-cadherin (1:100) (BD Bio transductions, 610182), Lgl (1:400) (*Sonawane et al., 2009*), PKC-ζ (C-20) (1:750) (Santa Cruz, sc-216), Chicken Anti GFP (1:200) (AbCAM, ab13970), Rabbit Anti- GFP (1:100) (Torrey Pines, TP401), Anti-RFP (1:200) (AbCAM, ab62341), β-catenin (P14L; 1:500) (*Schneider et al., 1996*), ZO-1 (1:100) (Invitrogen, 33–9100). The embryos were then washed five times in PBST and incubated with appropriate secondary antibodies Cy3 (1:750) (Jackson Immuno Research Labs, 115-165-144/146), Cy5 (1:750) (Jackson Immuno Research Labs, 115-175-144/146) or Alexa 488 (1:250) (Invitrogen, A-11029/A11034/A11039). The embryos were then washed five times with PBST and post fixed with 4%PFA in PBS for 30 min. The embryos were washed with PBS and serially upgraded in glycerol for imaging whole mounts.

## Immuno-histological tissue sections

After immunostaining and post fixation with 4% PFA, the embryos were washed twice with PBS and serially upgraded to anhydrous alcohol. The samples were then upgraded to Technovit-7100 + Hardener-1 (TV-1) (Electron Microscopy Sciences, #14653). The embryos were then transferred into

moulds filled with TV-1+ Hardener-2, were covered with a transparency/plastic sheet and left overnight at 37°C to polymerise.

The embryos in the polymerised blocks were cut transversely on Leica Microtome RM2265 to obtain the sections. The sections were picked on gelatine coated slides and mounted with Vectasheild Antifade Mounting Medium (Vector Laboratories, H-1000).

## Image acquisition

Images were acquired from the dorsal head epidermis of zebrafish embryos on Zeiss 510 LSM Meta, Zeiss LSM 710 or Zeiss LSM 880 confocal microscope with an EC Plan 63x/1.40 oil immersion objective lens at 1.5x optical zoom or using Zeiss 710 LSM confocal microscope with an APlanochromat 63x/1.40 oil immersion objective lens at 1.5x optical zoom and Z-step of 0.28 µm for fixed samples and 0.38 µm for live imaging. The images were taken at a resolution of 1024 × 1024. The pinhole was kept at 1 Airy Unit. The PMT gains were set using the genetic/developmental condition giving the brightest signal to prevent saturation, that is conditions were set using a) 72 hpf for *Figure 1*, *Figure 1—figure supplements 2*, *3*) WT siblings for *Figure 2*, *Figure 2—figure supplements 2* and *4*, and *Figure 3*) Ctrl MO in *Figures 4* and *5* for E-Cadherin and d) using E-cadherin MO in *Figure 5* for evaluating Lgl localisation. The PMT gains were kept constant within the experiments. Wherever essential, embryos having head epidermal clones injected with MO/mRNA and marked with GFP/mCherry, were sorted after immunostaining under mercury lamp in epifluorescence and analysed by confocal microscopy.

For live imaging, embryos of the specified genotype were injected at the 1 cell stage with 200 ng/µl *e-cadherin-mCherry/CAAX mCherry* mRNA. For imaging, 48 hpf embryos were immobilised in 0.8% low melting agarose (Sigma, A9414-100G). The embryos were screened on the basis of expression levels of E-cadherin. Animals which showed adequate signal for mCherry, with laser intensities kept between 2–4%, were subsequently imaged. The PMT Gains were kept constant throughout all the experimental sets.

To visualise the gradient of E-cadherin in a better manner in histological sections, the epidermal cells on top of the eye- where they are taller- were imaged (*Figure 1 B1, B2*, *Figure 1—figure supplement 1D*). For *has/apkc* mutants and wild-type siblings the sections were imaged at the dorsal head epidermis (*Figure 2—figure supplement 2A*).

## Quantification and statistical analysis

Fiji image analysis software (*Schindelin et al., 2012*) was used to quantify the E-cadherin, E-cadherin-mCherry or Lgl levels in the cells. For line plots, a line was drawn across the area marked and intensity profile was plotted using Fiji's inbuilt function. For quantifications across z-stacks, the cell boundaries were marked using line tool with a fixed width of 7 pixels to cover the cell membrane and the levels were measured at each z-level. For each condition, 4–6 non-neighbouring cells were quantified from each embryo. 7–10 embryos were quantified with minimum 30 cells per set. Note that quantification was done where the lateral domains were clearly visible. At the apical domain of the basal epidermis, neuronal axons interfere in quantifications. Therefore, such slices (typically 1–2 slices) were excluded from quantifications. At least three sets were quantified for each experiment unless mentioned otherwise. For *Figure 1* and *Figure 5*, two sets with minimum of 25 cells at each time point from each set or average 35 boundaries per category per set were quantified. For live imaging experiments, (*Figure 1—figure supplement 1*, and *Figure 2—figure supplement 4*) at least two sets with minimum of 29 cells were quantified. All the details of number of cells/boundaries/clones analysed for each experiment has been provided in *Figure 6—source data 1*. The data were then exported to R statistical software (*R Development Core Team, 2018*) on RStudio (*RStudio T, 2016*) for further analysis. The libraries ggplot2 (*Wickham, 2016*), dplyr (*Wickham et al., 2018*), tidyr (*Wickham and Henry, 2018*), purrr (*Henry and Wickham, 2018*), data. table (*Dowle et al., 2018*), scales (*Wickham, 2018a*), locfit (*Loader, 2013*; *Loader, 1999*), RCurl (*Lang and Team C, 2018*), stringr (*Wickham, 2018b*), gridExtra (*Baptiste and Antonov, 2017*), RColorBrewer (*Neuwirth, 2014*), and grid (*R Development Core Team, 2018*) were used for analysis.

Cell heights were calculated by counting the number of z-slices showing E-cadherin localisation. For comparison between different conditions for E-cadherin or Lgl, the cell heights were normalised

from 0 to 1, with 0 being apical and one being the basal, for each cell. Curves for Normalised Height versus E-cadherin levels were plotted for each cell. Each of these curves were then fitted using 'loc-fit' regression (*Loader, 2013*; *Loader, 1999*) in R statistical software, to interpolate E-cadherin and Lgl intensity values at fixed normalised heights of 0.1, 0.3, 0.5, 0.7 and 0.9 for each cell and thus get comparative values irrespective of the data acquisition points. These interpolated points were then used for comparisons between different genetic conditions.

For height-wise classification of peridermal cells of wild-type siblings and *has/apkc* mutants, distribution of the sibling cells was used as a reference. Cells were classified as 'Tall' when their heights were above the third quantile of this reference distribution, 'medium' when the cell heights were between the first and the third quantile, and 'short' when the heights were below the first quantile of the reference distribution as shown (*Figure 2—figure supplement 3B*).

In order to see the effect on the polarised distribution of E-cadherin, the maximum value of E-cadherin in each cell was normalised to 100% and the points were then plotted as a scatter plot. For better visualisation of points, the plot was binned in 15 bins on both the axes. Presence of a data point/s in the bin is represented as a coloured box. The number of data points in a bin is directly correlated to the darkness of the coloured boxes. More data points yielded darker boxes. To estimate the cells showing the abnormal E-cadherin distribution, a linear regression was fitted to the sibling data presented in graphs for aPKC sib (*Figure 2 A4*, *Figure 2—figure supplement 3 C2, D2*, *Figure 2—figure supplement 4D*). Regressions when fitted to the mutant data showed slopes and intercepts close to siblings, thus indicating the trends for both the genetic conditions are similar (not shown) but the noise levels may differ. The sibling data fitted with the regression line and a prediction interval of 99% is considered to encompass the baseline variability in siblings. Thus, data points lying outside this 99% prediction interval is classified as the noise in siblings. Since the sibling data provide the baseline for comparison, the noise in the mutant data was estimated as the mutant data points lying outside the 99% prediction interval of the sibling data. Any sibling or mutant peridermal cell having one or more data point outside 99% prediction interval was considered to have an abnormal distribution, contributing towards noise. The noise index was estimated as the percentage of the number of cells showing abnormal distribution.

For boundary wise quantifications for E-cadherin or Lgl levels, (presented in *Figure 4B,C,E,F*, *Figure 4—figure supplement 2*, *Figure 5 B5, B6, C5, C6*) the boundaries were divided into total six categories (*Figure 4—figure supplement 1 C1,C2*). To check for the layer autonomous effects, the boundaries were categorised as a) boundaries within the morphant clone cells (Clone-Clone or CC), b) boundaries between a clonal cell and a non-clonal cell (Non Clone- Clone or NC), and c) those between two non-clonal cells (Non Clone- Non Clone or NN). To check for the layer non autonomous effects, we categorised boundaries as d) those lying completely over/under a morphant clone (Complete Overlap or CO), e) or lay partially over/under the morphant clones (Partial Overlap or PO) and f) having no overlap with the morphant clones (No Overlap or NO) as shown in the schematic.

The statistical analysis was done using SigmaPlot 13 and R statistical software. Since most of the data failed Shapiro-Wilk Normality Test, non-parametric statistical test- Mann Whitney U test or Kruskal Wallis test with Dunn's post hoc test were performed. A p-value<0.05 was kept as a threshold for significance, in some cases threshold was extended till 0.08. Graphs were plotted in R and line plots were plotted in Fiji.

## Acknowledgements

We acknowledge TIFR-DAE and Wellcome-trust-DBT-India-Alliance (MS) for funding and Dr. Kalidas Kohale for zebrafish maintenance.

## Additional information

### Funding

| Funder | Grant reference number | Author |
| --- | --- | --- |
| Wellcome Trust/DBT India Alliance | 500129-Z-09-Z | Mahendra Sonawane |

| Tata Institute of Fundamental Research | 12P121 | Mahendra Sonawane |
|---|---|---|

The funders had no role in study design, data collection and interpretation, or the decision to submit the work for publication.

## Author contributions
Prateek Arora, Software, Formal analysis, Investigation; Shivali Dongre, Formal analysis, Validation, Investigation; Renuka Raman, Formal analysis, Investigation; Mahendra Sonawane, Conceptualization, Supervision, Funding acquisition

## Author ORCIDs
Prateek Arora http://orcid.org/0000-0003-0822-9240
Renuka Raman http://orcid.org/0000-0002-9723-9442
Mahendra Sonawane https://orcid.org/0000-0002-1749-6247

## Ethics
Animal experimentation: The zebrafish maintenance and experimental protocols used in this study were approved by the institutional animal ethics committee vide Reference- TIFR/IAEC/2017-11.

## Decision letter and Author response
Decision letter https://doi.org/10.7554/eLife.49064.sa1
Author response https://doi.org/10.7554/eLife.49064.sa2

## Additional files

### Supplementary files
• Supplementary file 1. R scripts consisting of codes used to analyse data for localisation of E-cadherin and Lgl, and for cell morphological parameters from various genetic conditions.

• Transparent reporting form

### Data availability
All data generated or analysed during this study are included in the manuscript and supporting files. Source data files have been provided for Figures 1, 2, 3, 4, 5, Figure 2- figure supplement 3 and 4, Figure 4- figure supplement 2.

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
