## [Decision Letter]

Thank you for sending your article entitled "Stepwise polarisation of developing bilayered epidermis is mediated by aPKC and E-cadherin in zebrafish" for peer review at *eLife*. Your article has been evaluated by three peer reviewers, one of whom is a member of our Board of Reviewing Editors, and the evaluation has been overseen by Didier Stainier as the Senior Editor.

This paper breaks new ground into how apicobasal polarity is established in the developing epidermis using the zebrafish as a model. While we all felt that the paper was of fundamental interest to our larger understanding of how apical-basal polarity is established in developing epidermis, all of the reviewers in their reviews and discussion had concerns about the basic localization of E-cadherin. This was a concern for the wild type samples--as they differed between different figures, which made interpretation of modulations with different mutants difficult to interpret. We all felt that all E-cadherin localization studies as well as other localizations needed to be shown within the context of other clear markers. Without the comparisons, it is difficult to assess where proteins are localizing (a sort of chicken and egg conundrum). Even having an overall membrane marker like DiI and DNA stain might be useful backdrops in which to make these comparisons as well as scans going through several cell layers, so that one could see E-cadherin basally in the top layer and apically in the bottom layer. Other points raised by the specific comments of reviewers (below) question the causality of the E-cadherin experiments and would also need to be addressed.

Reviewer #1:

The manuscript by Arora and colleagues demonstrates a novel mechanism for how epidermal cells within the zebrafish establish apical basal polarity during development, which could have implications for other bilayered epithelia that coat organs. They show that, surprisingly, E-cadherin is polarized more basally in the outer peridermal cells and apical in the basal cells. Using morpholinos and E-cadherin rescue that E-cadherin in one layer sets up polarity in the overlying or underlying layer. Using mutations in lgl and has (apkc), they show that E-cadherin and apkc restrict localization of each in a cell-autonomous way and that E-cadherin negatively regulates Lgl localization. This is a nice study that shows how polarity may be set up in multilayered epithelia that by inference of how cell-cell contacts are set up and reinforced in a monolayer extrapolates a novel function in a bilayer for setting up polarity in the layer below or above. I think that this study is suitable for *eLife* but I have some issues that I would like the authors to address before this happens.

1) The manuscript needs tightening up. It is written in a long-winded way with too much detail in the main text where I found it difficult to follow the main directive of each experiment. Likewise, the figures could be more compact-there were many times where I wanted to see several of the markers that they were looking at co-stained, rather than in supplementary figures at the bottom that lacked the other marker context. I think it is imperative that the co-stainings are together, especially as the localization at the apex or base is being measured.

2) The apical versus basal localization needs better analysis and description. Everyone assumes that E-cadherin localizes apically so it is still hard to believe that most of it is basal in the periderm. To prove this better, they need to (a) label with another marker in the same cells so that we can be certain that we are actually starting at the top of the cell and scanning to the very bottom of the cell, (b) indicate in μm where we are from the top of the cell, (c) show a scan going from the top of a peridermal cell to the bottom of the basal cell. I think that these experiments would prove their point better as it currently seems very arbitrary what they call top and bottom. Again, this study could be combined with the other markers (b-catenin, lgl, apkc) in Figure 1, which could correlate well with the model in Figure 6. As it is now, we have to piece together the bits of data and it isn't obvious or all that convincing.

3) Why is there more E-cadherin apically in Figure 2 wildtype than in Figure 1? In Figure 1 there is hardly any E-cadherin and that just seems weird.

Reviewer #2:

This paper describes how the epidermis polarizes in zebrafish. The authors propose that aPKC promoted E-cadherin polarization in periderm affects the levels of E-cadherin and Lgl2 in the underlying basal epidermal cell layer. The mutant phenotypes are very carefully analysed and the reduction of E-cadherin in a layer non-autonomous manner is an interesting observation. However, I believe there are serious issues that need to be addressed.

1) The authors observe a reversed trend for E-cadherin polarity in periderm and underlying basal epidermal cell layers. How can this observation be reconciled with the non-reversed apicobasal polarities of these two layers?

2) The authors suggest a novel stepwise mechanism of polarity transduction. However, the lasting polarity of basal epidermal cells in has/apkc mutants might support an independent mechanism of polarization in this layer. Moreover, the data on E-cadherin knock-down do not clearly support an E-cadherin mediated transduction of polarity mechanism. E-cadherin knock down experiments should be better quantified, just like the mutant phenotypes. aPKC, E-cadherin and Lgl2 polarity distributions with reduced levels of E-cadherin should be provided. Only this will allow to conclude on the role of E-cadherin trans-binding in cell polarization.

3) Trans-interactions of E-cadherin is the suggested mediator of stepwise polarity transduction. However, a mechanistic insight into this process is lacking. The authors should come up at least with some preliminary insight into the mechanism(s) by which E-cadherin trans-binding controls the level and distribution of E-caherin in the binding cells.

Reviewer #3:

In this paper the authors study a very important question, which is how the polarized distribution of adherens junctions in the developing epidermis is achieved. Specifically, they show that in zebrafish embryos the periderm and basal epidermis display an opposite gradient of E-cadherin expression and state a role of aPKC in maintaining this E-cadherin gradient in the periderm. Furthermore, they propose that peridermal polarity, initiated by aPKC, is transduced in a stepwise manner by E-cadherin to the basal layer in a non-autonomous manner.

Essential revisions:

1) All the results rely exclusively on the quantification of fluorescence intensity in several cell boundaries, at each z-point, manually applying Fiji software tools. These values are then converted to AU. However, no other means supporting the use of this approach are provided. I think it is difficult to use measurements of fluorescence intensity of antibody staining for comparisons. Can the authors provide other experiments to support the exclusive use of this approach?

2) Authors say that adherens junctions are localized in a polarized manner. However, there is a discrepancy: they show that E-cadherin is faintly present at 48hpf in the basal domain of the periderm, whereas expression of β-catenin can be observed since 24hpf. How can this be explained? What is β-catenin attached to? Can this unveil one of the drawbacks pointed in before?

3) The "length" of the apico-basal gradient and cell sizes are measured according to the number of z-stacks that show the specific staining. Authors should provide the z-stacks projections to back up all the performed quantifications. With this information they will be able to observe at what apico-basal level aPKC disappears, and E-cadherin expression appears. One possibility is to use the transgenic line Lyn:GFP, and immunostain for E-cadherin and aPKC in that context. This would help to understand what happens with the height of the cells. According to the Figure 2B1 it seems that the mutant cells are indeed smaller than the wt siblings (in the very same square surface 4 wt basal epidermis wt cells can fit, and around 6 has/apkc cells are included), although the apical perimeter measurements do not show so. Are the authors sure that all pictures have the same magnification? Are the cells smaller but have more membrane blebblings that could account for these numbers? Shouldn't the numbers be normalized according to overall membrane staining, not to E-cadherin expression?

4) The functional experiments do not provide convincing data to support the stated conclusions. Again, all the analyses rely on the approaches mentioned before, and they are difficult to interpret. How aPKC, lgr and E-cadherin are distributed along the apico-basal domain? This would be important to better understand the phenotypes.

5) Controls about the efficiency of the morpholino should be included, as well as numbers of embryos displaying the phenotype. It would be nice to have more information about the kind of morpholino employed.

Can authors provide the numbers for wt/mutants? How many cells/boundaries were analyzed and how many embryos displayed the phenotype? Can they provide information about the penetrance and the expressivity of the observed phenotypes?

---

## [Author Response]

While we all felt that the paper was of fundamental interest to our larger understanding of how apical-basal polarity is established in developing epidermis, all of the reviewers in their reviews and discussion had concerns about the basic localization of E-cadherin. This was a concern for the wild type samples--as they differed between different figures, which made interpretation of modulations with different mutants difficult to interpret.

We concur with the reviewers’ concern. We would like to point out that the study involves fluorescence intensity measurements and it was important to maintain the fluorescence intensities at non-saturating levels while imaging. The intensity differences seen in some of our samples are a consequence of our attempt to control the saturation. The E-cadherin levels are high at 72hpf and show saturation while imaging if the gain is set according to the 48hpf embryos. As a consequence, in these sets wherein the PMT gain is set using 72hpf, the E-cadherin intensity at 48hpf seems much weaker in comparison to the other sets. This is obvious when Ecadherin levels at 48hpf in Figure 1A are compared with those in Figure 2. We have included these imaging details in the Materials and methods section in the revised version.

We all felt that all E-cadherin localization studies as well as other localizations needed to be shown within the context of other clear markers. Without the comparisons, it is difficult to assess where proteins are localizing (a sort of chicken and egg conundrum). Even having an overall membrane marker like DiI and DNA stain might be useful backdrops in which to make these comparisons as well as scans going through several cell layers, so that one could see E-cadherin basally in the top layer and apically in the bottom layer.

This was a very good suggestion. To address this issue, we tried various membrane labels such as CAAX-GFP, CAAX-mCherry, DiI, FM 1-43 dye, WGA, and Tg(bactin:mKate2-ras) (Sidhaye and Norden, 2017) to mark the membrane in both the epidermal layers. However, only Lyn-GFP driven by *claudin b* promoter (suggested by reviewer # 3) in peridermal cells worked the best for confocal microscopy as well as immunohistology. In the revised version, we have included the histology sections (imaged on lateral head region or on the dorsal head) showing E-cadherin localisation with respect to the membrane marker, Lgl and aPKC. In addition, we show full z-stack images across the two layers.

Other points raised by the specific comments of reviewers (below) question the causality of the E-cadherin experiments and would also need to be addressed.

In the revised version, we have assessed the E-cadherin phenotypes by quantifying E-cadherin, Lgl and aPKC levels. To obtain preliminary mechanistic insights into the functioning of E-cadherin in regulation of polarity, we attempted to test whether the function of E-cadherin as a mediator of adhesion or its role in organisation of cortex/signalling is important for regulation of polarity across the layers. For this, we generated mutant version of E-cadherin lacking a) EC-1 and EC-2 domains (Kardash et al., 2010; Leckband and de Rooij, 2014) and b) the cytoplasmic tail of Ecadherin, which harbours β-catenin binding domain (Gottardi et al., 2001). Unfortunately, both these deletion mutants did not localise when mRNA was injected, precluding the possibility of assessing them in E-cadherin deficient background (by morpholino injections). Also, upon plasmid injections, only EC1-EC2 domain deletion localised to the membrane. Our data indicate the EC1EC2 domain, presumably via tension-mediated mechanisms, regulate Lgl localisation in the epidermis. This preliminary analysis has been included in the revised version.

Reviewer #1:The manuscript by Arora and colleagues demonstrates a novel mechanism for how epidermal cells within the zebrafish establish apical basal polarity during development, which could have implications for other bilayered epithelia that coat organs. They show that, surprisingly, E-cadherin is polarized more basally in the outer peridermal cells and apical in the basal cells. Using morpholinos and E-cadherin rescue that E-cadherin in one layer sets up polarity in the overlying or underlying layer. Using mutations in lgl and has (apkc), they show that E-cadherin and apkc restrict localization of each in a cell-autonomous way and that E-cadherin negatively regulates Lgl localization. This is a nice study that shows how polarity may be set up in multilayered epithelia that by inference of how cell-cell contacts are set up and reinforced in a monolayer extrapolates a novel function in a bilayer for setting up polarity in the layer below or above. I think that this study is suitable for eLife but I have some issues that I would like the authors to address before this happens.1) The manuscript needs tightening up. It is written in a long-winded way with too much detail in the main text where I found it difficult to follow the main directive of each experiment. Likewise, the figures could be more compact-there were many times where I wanted to see several of the markers that they were looking at co-stained, rather than in supplementary figures at the bottom that lacked the other marker context. I think it is imperative that the co-stainings are together, especially as the localization at the apex or base is being measured.

We have tightened up the manuscript by either eliminating the unimportant details or by moving them to the methods part. We have also revised the Figure 1 and Figure 1—figure supplement 1 to include all the co-staining together for the ready reference of the readers.

2) The apical versus basal localization needs better analysis and description. Everyone assumes that E-cadherin localizes apically so it is still hard to believe that most of it is basal in the periderm. To prove this better, they need to (a) label with another marker in the same cells so that we can be certain that we are actually starting at the top of the cell and scanning to the very bottom of the cell, (b) indicate in μm where we are from the top of the cell, (c) show a scan going from the top of a peridermal cell to the bottom of the basal cell. I think that these experiments would prove their point better as it currently seems very arbitrary what they call top and bottom. Again, this study could be combined with the other markers (b-catenin, lgl, apkc) in Figure 1, which could correlate well with the model in Figure 6. As it is now, we have to piece together the bits of data and it isn't obvious or all that convincing.

This was a very useful suggestion. To address this comment, we tried several membrane markers such as CAAX-GFP, CAAX-mCherry, DiI, FM 1-43 dye, Tg(bactin:mKate2-ras) (Sidhaye and Norden, 2017) and WGA to mark the membrane in both the epidermal layers. As stated above Lyn-GFP driven in peridermal cells by *claudin b* promoter (suggested by reviewer # 3) worked the best. We have now provided histological sections as well as confocal scans going through the entire epidermis (top of the periderm to the bottom of the basal epidermis) showing localization of E-cadherin, aPKC and Lgl in the context of Lyn-GFP. These data are included in revised Figure 1 and Figure 1—figure supplement 1. We have also mentioned the actual distance (in µm) of the confocal slice from the top

3) Why is there more E-cadherin apically in Figure 2 wildtype than in Figure 1? In Figure 1 there is hardly any E-cadherin and that just seems weird.

We concur with the reviewers’ concern. As stated in response to the Editor’s comment, the intensity differences are a consequence of our attempt to avoid the saturation. E-cadherin levels are high at 72hpf and show saturation while imaging if the PMT gain is set according to the 48hpf embryos. Therefore, in sets wherein the PMT gain is set using 72hpf, the E-cadherin intensity at 48hpf seems much weaker in comparison to the other sets. We have included imaging details in the Materials and methods section in the revised version.

Reviewer #2:This paper describes how the epidermis polarizes in zebrafish. The authors propose that aPKC promoted E-cadherin polarization in periderm affects the levels of E-cadherin and Lgl2 in the underlying basal epidermal cell layer. The mutant phenotypes are very carefully analyzed and the reduction of E-cadherin in a layer non-autonomous manner is an interesting observation. However, I believe there are serious issues that need to be addressed.1) The authors observe a reversed trend for E-cadherin polarity in periderm and underlying basal epidermal cell layers. How can this observation be reconciled with the non-reversed apicobasal polarities of these two layers?

This is an interesting question. Our data clearly points to the fact that E-cadherin polarity is reversed in the epidermal layers. While there is basolateral enrichment of E-cadherin in the periderm, its localisation in the basal epidermis is clearly enriched at the apico-lateral domain. However, we did not observe localisation of aPKC in the apical domain of the basal epidermal cells during early development. This means that although there is a histologically defined apical domain (opposite to the basal domain facing the ECM) for the basal cells, this domain may not be molecularly specified at this early stage. It appears that the molecular identity of the histological apical domain of the basal epidermal cells remains undefined at the early stages of the epidermis development (or it may be a feature of the bilayered epithelial systems in general). E-cadherin sets up the rudimentary but reversed polarities in the periderm and basal epidermis, making apical domain of the periderm (aPKC localisation, microridges) and basal domain of the basal epidermis (hemidesmosome formation), more specified. The basal epidermal cells may get the real apical identity at the “molecular” level possibly at a later stage when the polarity of the basal cells becomes important for the asymmetric cell division during stratification, as shown in mice (Helfrich et al., 2007; Lechler and Fuchs, 2005; Niessen et al., 2013; Tunggal et al., 2005; Williams et al., 2011, 2014). Experimental reconciliation of the reversed E-cadherin polarity with the non-reversed apical-basal polarities require much detailed analyses involving multiple apicalbasal markers at early as well as later developmental stages. Such analyses are not feasible at this stage. Therefore, we have discussed this issue in detail in the revised version.

2) The authors suggest a novel stepwise mechanism of polarity transduction. However, the lasting polarity of basal epidermal cells in has/apkc mutants might support an independent mechanism of polarization in this layer. Moreover, the data on E-cadherin knock-down do not clearly support an E-cadherin mediated transduction of polarity mechanism. E-cadherin knock down experiments should be better quantified, just like the mutant phenotypes. aPKC, E-cadherin and Lgl2 polarity distributions with reduced levels of E-cadherin should be provided. Only this will allow to conclude on the role of E-cadherin trans-binding in cell polarization.

In the revised version, we have quantified E-cadherin, Lgl and aPKC levels to measure the effect on their localization upon *e-cadherin* knockdown. These analyses clearly reveal the layerautonomous as well as layer non-autonomous effect of E-cadherin loss in the epidermis.

As for the lack of polarity phenotype in the basal epidermis of the *has/apkc* mutant – – it is important to note that in the absence of aPKC function, the E-cadherin polarity is noisy but not completely lost. Such a partially polarised periderm may still be able to transduce the polarity cues to the basal epidermis to set up the E-cadherin polarity. However, in the absence of E-cadherin in the periderm, such polarity cues are completely lost resulting in a loss of E-cadherin localisation and increased Lgl localisation in the basal layer. It is likely that there are redundant mechanisms controlling E-cadherin polarity in the periderm. To fully perturb the E-cadherin polarity, disruption of these redundant pathway components would be essential. These arguments have been included in the revised version of the manuscript.

3) Trans-interactions of E-cadherin is the suggested mediator of stepwise polarity transduction. However, a mechanistic insight into this process is lacking. The authors should come up at least with some preliminary insight into the mechanism(s) by which E-cadherin trans-binding controls the level and distribution of E-caherin in the binding cells.

We made an attempt to get mechanistic insights using mutant version of E-cadherin lacking EC-1 and EC-2 domains (Kardash et al., 2010; Leckband and de Rooij, 2014) and cytoplasmic tail of E-cadherin, which harbours β-catenin binding domain (Gottardi et al., 2001). This approach did not yield clear results. Both the deleted versions did not localise when mRNA was injected. This did not allow us to test their phenotypes in the absence of endogenous E-cadherin (achieved by morpholino injections). Besides, plasmid encoded cytoplasmic tail deletion mutant did not localise to the membrane in our hands. EC1-EC2 domain deleted version of E-cadherin localised to the membrane when expressed under CMV promoter using plasmid. Our data suggest that EC1-EC2 domains are important for regulation of Lgl localisation. However, expression of this mutant version did not have any influence on E-cadherin localisation either autonomously or in layer-non-autonomous manner. These data suggest that the effect is not mediated via homophilic interactions but tension-mediated mechanisms are at play in this regulation. This is also supported by the fact that loss of E-cadherin results in much wide-spread (beyond the clones) phenotypes in E-cadherin and Lgl2 localisation.

Reviewer #3:In this paper the authors study a very important question, which is how the polarized distribution of adherens junctions in the developing epidermis is achieved. Specifically, they show that in zebrafish embryos the periderm and basal epidermis display an opposite gradient of E-cadherin expression and state a role of aPKC in maintaining this E-cadherin gradient in the periderm. Furthermore, they propose that peridermal polarity, initiated by aPKC, is transduced in a stepwise manner by E-cadherin to the basal layer in a non-autonomous manner.Essential revisions:1) All the results rely exclusively on the quantification of fluorescence intensity in several cell boundaries, at each z-point, manually applying Fiji software tools. These values are then converted to AU. However, no other means supporting the use of this approach are provided. I think it is difficult to use measurements of fluorescence intensity of antibody staining for comparisons. Can the authors provide other experiments to support the exclusive use of this approach?

Currently, quantification of fluorescence intensity is the best possible way for us to show the Ecadherin polarity, at the endogenous levels, and assess the effect of *apkc* loss on it. We have done this reproducibly in the lab and others have also used similar approaches (Bradley and Drissi, 2011; Hultin et al., 2017; Mantyh et al., 1997; Pocha et al., 2011; Shi et al., 2003). To address the reviewers’ concern, we have included live imaging based quantification of mCherry tagged E-cadherin. These analyses further support our claim of reverse polarity of E-cadherin in the epidermal layers of zebrafish (Figure 1—figure supplement 4) as well as the noisy polarity phenotype that we observe in *has/apkc* mutant (Figure 2—figure supplement 4).

In addition, we have included histological sections to show the polarised localisation of Ecadherin. We have also analysed localisation of E-cadherin, Lgl2 and aPKC in the presence of a membrane marker (lyn-GFP) using immunohistology. Our data now directly reveal the E-cadherin polarity in wild-type and that the levels of E-cadherin decrease in *has/apkc* mutant embryos.

2) Authors say that adherens junctions are localized in a polarized manner. However, there is a discrepancy: they show that E-cadherin is faintly present at 48hpf in the basal domain of the periderm, whereas expression of β-catenin can be observed since 24hpf. How can this be explained? What is β-catenin attached to? Can this unveil one of the drawbacks pointed in before?

We have shown that the depletion of E-cadherin results in a considerable reduction in Betacatenin localisation (Figure 5—figure supplement 1). This result indicates that there are no other cadherins, (for example, P-cadherin) present in the epidermis, to which Β-catenin binds. We have used cross-reacting primary antibodies raised against mouse E-cadherin and chicken Betacatenin. The binding affinities of these cross-reacting antibodies towards the zebrafish antigen may not necessarily be the same. This would explain the intensity difference seen between Ecadherin and Β-catenin staining.

3) The "length" of the apico-basal gradient and cell sizes are measured according to the number of z-stacks that show the specific staining. Authors should provide the z-stacks projections to back up all the performed quantifications. With this information they will be able to observe at what apico-basal level aPKC disappears, and E-cadherin expression appears. One possibility is to use the transgenic line Lyn:GFP, and immunostain for E-cadherin and aPKC in that context. This would help to understand what happens with the height of the cells.

This was a very useful suggestion. We have performed localization analysis of E-cadherin, aPKC and Lgl2 in Lyn-GFP background. This analysis – done by confocal microscopy as well as immunohistology – – is included in the revised version. Confocal z-stacks projections throughout the epidermis have also been included in the revised version. Figure 1A1, Figure 1—figure supplement 1 and Figure 1—video 1 show that aPKC level decrease and E-cadherin level increase along the apico-basal axis.

According to the Figure 2B1 it seems that the mutant cells are indeed smaller than the wt siblings (in the very same square surface 4 wt basal epidermis wt cells can fit, and around 6 has/apkc cells are included), although the apical perimeter measurements do not show so. Are the authors sure that all pictures have the same magnification? Are the cells smaller but have more membrane blebblings that could account for these numbers?

We thank the reviewer for pointing this out. In the revised version, we have presented a better representative image.

Shouldn't the numbers be normalized according to overall membrane staining, not to E-cadherin expression?

As discussed earlier, we have used a membrane marker (lyn-GFP) along with E-cadherin localisation in the revised version, which is useful as a reference for the qualitative assessment. We also show reduction in E-cadherin localisation in the *has/apkc* mutant in presence of lyn-GFP by immunohistology as well as by confocal microscopy. However, re-quantification and normalisation using a membrane marker, across all the wild type stages and genetic conditions is not feasible within a short time frame.

4) The functional experiments do not provide convincing data to support the stated conclusions. Again, all the analyses rely on the approaches mentioned before, and they are difficult to interpret. How aPKC, lgr and E-cadherin are distributed along the apico-basal domain? This would be important to better understand the phenotypes.

We have quantified the Lgl, E-cadherin and aPKC levels in absence of *e-cadherin* function. These quantifications support our claim that E-cadherin controls layer non-autonomous localization of E-cadherin, and layer autonomous as well as layer non-autonomous localization of Lgl2

5) Controls about the efficiency of the morpholino should be included, as well as numbers of embryos displaying the phenotype. It would be nice to have more information about the kind of morpholino employed.Can authors provide the numbers for wt/mutants? How many cells/boundaries were analyzed and how many embryos displayed the phenotype? Can they provide information about the penetrance and the expressivity of the observed phenotypes?

Details about type of morpholino, the penetrance of the phenotype, number of wt/mutant embryos analysed, cell/boundary numbers have been provided in the Results section, Discussion section and Materials and methods section and source data files. In the revised version, we have also compiled this information at one place (Figure 6—source data 1).